# CobWeb 1.0: Machine Learning Tool Box for Tomographic Imaging

Swarup Chauhan[1,2], Kathleen Sell[1,3,*], Wolfram Rühaak[4], Thorsten Wille[5], Ingo Sass[2]

[1] Institute for Geosciences, Johannes Gutenberg-University, Mainz 55099, Germany
[2] Institute of Applied Geosciences, University of Technology, Darmstadt 64287, Germany
[3] igem – Institute for Geothermal Resource Management, Berlinstr. 107a, Bingen 55411, Germany
[4] Bundesgesellschaft für Endlagerung mbH (BGE), Peine 38226, Germany
[5] APS Antriebs-, Prüf- und Steuertechnik GmbH, Götzenbreite 12, Göttingen-Rosdorf 37124, Germany

* now at: Ministry of Economic Affairs Rhineland Palatine, Stiftsstrasse 9, Mainz 55116, Germany

*Correspondence to: Swarup Chauhan (ssschauhan@gmail.com)

**Abstract.**

Despite the availability of both commercial and open-source software, an ideal tool for digital rock physics analysis for accurate automatic image analysis at ambient computational performance is difficult to pinpoint. More often image segmentation is driven manually where the performance remains limited to two phases. Discrepancies due to artefacts cause inaccuracies in image analysis. To overcome these problems, we have developed CobWeb 1.0 which is automated and explicitly tailored for accurate grayscale (multi-phase) image segmentation using unsupervised and supervised machine learning techniques. In this study, we demonstrate image segmentation using unsupervised machine learning techniques. The simple and intuitive layout of the graphical user interface enables easy access to perform Image enhancement, Image segmentation and further to obtain the accuracy of different segmented classes. The graphical user interface enables not only processing of a full 3D digital rock dataset but also provides a quick and easy region-of-interest selection, where a representative elementary volume can be extracted and processed. The CobWeb software package covers image processing and machine learning libraries of MATLAB® used for image enhancement and image segmentation operations, which are compiled into series of windows executable binaries. Segmentation can be performed using unsupervised, supervised and ensemble classification tools. Additionally, based on the segmented phases, geometrical parameters such as pore size distribution, relative porosity trends and volume fraction can be calculated and visualized. The CobWeb software allows the export of data to various formats such as ParaView (.vtk), DSI Studio (.fib) for visualization and animation and Microsoft® Excel and MATLAB® for numerical calculation and simulations. The capability of this new software is verified using high-resolution synchrotron tomography datasets, as well as lab-based (cone-beam) X-ray micro-tomography datasets. Albeit the high spatial resolution (sub-micrometer), the synchrotron dataset contained edge enhancement artefacts which were eliminated using a novel dual filtering and dual segmentation procedure.

## 1. Introduction

Currently, a vast number of available commercial and open-source software packages for pore-scale analysis and modelling exists (compiled in Figure 1), but dedicated approaches to verify the accuracy of the segmented phases are lacking. To the best of our knowledge, the current practice among researchers is to alternate between different available software tools and to synthesize the different datasets using individually aligned workflows. Porosity and in particular, permeability can vary dramatically with small changes in segmentation, as significant features on the pore-scale get lost when thresholding greyscale tomography images to binary images, even if using the most advanced data acquiring techniques like synchrotron tomography (Leu et al., 2014). Our new CobWeb 1.0 visualization and image analysis toolkit addresses some of the challenges of selecting representative elementary volume (REV) for X-ray computed tomography (XCT) datasets reported earlier by several researchers (Zhang D et al., 2000; Gitman et al., 2006; Razavi et al., 2007; Al-Raoush and Papadopoulos, 2010; Costanza-Robinson et al., 2011; Leu et al., 2014). The software is built on scientific studies which have been peer-reviewed and accepted in the scientific community (Chauhan et al., 2016b; Chauhan et al., 2016a). The spinoff for these studies was not the lack of accuracy provided by manual segmentation schemes, but the subjective assessment and non-comparability caused by the individual human assessor. Therefore, automated segmentation schemes offer speed, accuracy and possibility to intercompare results, enhancing traceability and reproducibility in the evaluation process. To our knowledge, none of the XCT software used in rock science community relies explicitly on machine learning to perform segmentation, which makes the software unique.

Despite many review articles and scientific publication highlight potential of machine learning and deep learning (Iassonov et al., 2009; Cnudde and Boone, 2013; Schlüter et al., 2014), software libraries or toolbox are seldom made available. Thus, with CobWeb we started for the first time to fill this gap, and despite its limited volume rendering capabilities— it is a useful tool and current version of the software can be applied in scientific and industrial studies. CobWeb provides an appropriate test platform, where new segmentation and filtration schemes can be tested and used as a complementary tool to the simulation software GeoDict and Volume Graphics. The simulation software (GeoDict and Volume Graphics) have benchmarked solvers for performing flow, diffusion, dispersion, advection type simulation, but their accuracy relies heavily on the finely segmented datasets. This software is based on a machine learning approach with great potential for segmentation analysis as introduced previously (Chauhan et al., 2016b; Chauhan et al., 2016a). Further, this software tool package was developed on a MATLAB® workbench and can be used as a Windows stand-alone executable (.exe) files or as a MATLAB® plugin. The dataset for the gas hydrate sediment (GH) geomaterials was acquired using monochromatic synchrotron X-ray, unhampered by beam hardening; (Sell et al., 2016) highlighted problems with edge enhancement artefact and recommended image morphological strategies to tackle this challenge. In this paper, we describe therefore also a strategy to eliminate ED artefacts using the same dataset but applying the new machine learning approach.

## 2. Image Processing

Image pre-processing is one of the essential and precautionary steps before image segmentation (Iassonov et al., 2009; Schlüter et al., 2014). Image enhancement filtering techniques help to reduce artefacts such as blur, background intensity and contrast variation. Whereas, denoise filter such as median filter, non-local means filter, and anisotropic diffusion filter can assist in lowering the phase misclassification and improving the convergence rate of automatic segmentation schemes. CobWeb 1.0 is equipped with image enhancement and denoise filters, namely, *imsharpen, non-local-means, anisotropic diffusion and fspecial*, which are commonly used in the XCT image analysis community.

### 2.1.1. Imsharpen Image Enhancement

Despite at the instrument level, different measures can be taken to improve the resolution of the X-ray volumetric data, the contrast, in the XCT images depends particularly on the composition and corresponding densities (optical depth) of the test sample. Therefore, it is somewhat difficult to enhance contrast at the experimental setup or at the x-rays system design control stage. Thus, the contrast needs to be enhanced or adjusted after the volumetric image has been generated. For this purpose, image sharpening can be used. Image sharpening is a sort of contrast enhancements. The contrast enhancements generally take places at the contours, were high and low greyscale pixel values intensities meet (Parker, 2010).

### 2.1.2. Anisotropic Diffusion Image Filtering

For intuition purposes, Anisotropic Diffusion filter (AD) can be thought as (Gaussian) blur filter. AD blurs the image, where it carefully smooths the textures in the image by preserving its edges (Kaestner et al., 2008; Porter Mark L. et al., 2010; Schlüter et al., 2014). To achieve the smoothing along with edge preservation, the AD filter performs an iteration to solve non-linear partial differential equations (PDE) of diffusion:

$$\frac{\partial I}{\partial t} = c(x, y, t)\Delta I + \nabla c \,.\, \nabla I \qquad (1)$$

Where,

$I$ is the image, $t$ is the time of evolution and $c$ is the flux which controls the rate of diffusion at any point in the image.

(P. Perona and J. Malik, 1990) introduce a flux function $c$ to follow an image gradient and stop or restrain the diffusion when it reaches the region boundaries (edges preservation).

Given by

$$c(\|\nabla I\|) = e^{-(\|\nabla I\|\,\kappa)^2} \qquad (2)$$

$$(\|\nabla I\|) = \frac{1}{1+(\frac{\|\nabla I\|}{\kappa})^2} \qquad (3)$$

Here, the parameter $\kappa$ is a tuning parameter that determines if the given edge to be considered as a boundary or not. A large value of $\kappa$ lead to an isotropic solution and the edges are removed. For our investigations the parameter $\kappa$ (threshold stop) was

fixed to the value 22 968, which is the edge preservation limit between quartz grain and hydrate phase. The desired denoising (blurring/smoothing) was achieved within five iteration steps.

### 2.1.3. Nonlocal Means Image Filtering

Nonlocal means (NLM) filter is based on the assumption that the image contains an extensive amount of self-similarity (A. Buades et al., 2005; Shreyamsha Kumar, 2013). Based on this assumption (A. Buades et al., 2005) extended the linear neighbourhood SUSAN filter (Smith and Brady, 1997) with nonlocal class. Thus, through the nonlocal class, the spatial search for similar pixel values is not restricted to a constrained neighbourhood pixel but the whole image is part of the search for similar pixel values. Given by the equation

$$NL(i) = \sum_{j \in I} w(i,j)v(j) \qquad (4)$$

Where,

$NL(i)$ is the estimated nonlocal intensity of the pixel $i$

$I$ is the image

$w(i,j)$ is the weight (or average value) applied to noisy image $v(j)$ to obtain and restore the pixel $i$.

However, for a practical and computational reason, the search is performed within a search window or neighbourhood patches, and $w(i,j)$ evaluates similarity in pixel intensities between local neighbourhood patches. Where the weight $w(i,j)$ is calculated as

$$w(i,j) = \frac{1}{Z(i)} \, e^{-\frac{\left\| v(N_i) - v(N_j) \right\|^2_{2,\sigma}}{h^2}} \qquad (5)$$

Where. $Z(i)$ is a normalization constant

$$Z(i) = \sum_j e^{-\frac{\left\| v(N_i) - v(N_j) \right\|^2_{2,\sigma}}{h^2}} \qquad (6)$$

$v(N_i)$ , $v(N_j)$ are the local neighbourhood patches.

The similarity is fulfilled as the Euclidean distance between the local neighbourhoods patches exponentially decreases.

$\sigma > 0$ is the standard deviation.

In eq. (5) and (6), the distance function $\left\| v(N_i) - v(N_j) \right\|^2$ is pointwise multiplied (convolved) with $\sigma$, to ensure a fair contribution of pixel values to the weighted function.

### 2.1.4. fspecial Image Filtering

fspecial helps in creating a 2D high pass and low pass filters. High pass filters are used for sharpening and edge detection, whereas low pass filters are used for smoothing the image quality. Frequently used high pass filters are Laplacian and Sobel masks (kernel), and most often used low pass filter is the Gaussian smoothing mask (mask). However, in n the current version

of CobWeb, fspecial is implemented as an averaging filter. The filter is directly applied on the 2D slices without any convolution with the filter kernel.

## 2.2. Image Segmentation

A digital image comprises pixels of colour or greyscale intensities. Image segmentation is partitioning or classification the pixel intensities into disjoint regions that are homogenous with respect to some characteristics (Bishop, 2006). There are continuous research efforts done in various international groups to improve and developed image segmentation approaches (Mjolsness and DeCoste, 2001). In particular, the most popular and relevant image segmentation approaches for analyzing X-ray tomographic rock images are presented in the review studies done by (Iassonov et al., 2009) and (Schlüter et al., 2014). We use machine learning techniques for image segmentation and have implemented algorithms such as K-means, Fuzzy C-means (unsupervised), least-square support vector machine (LSSVM) (supervised), bragging and boosting (ensemble classifiers) for automatic segmentation Chauhan et al., (2016a), (Chauhan et al., 2016b) and references within. The performance of these machine learning technique can be assessed by matrices such as entropy, receiver operational characteristics (ROC), 10-fold cross-validation (Chauhan et al., 2016b). Below all the above-mentioned algorithms are described in brief.

### 2.2.1. Unsupervised Machine Learning Techniques

K-means is one of the simplest, yet, robust unsupervised machine learning (ML) algorithms commonly used in partitioning data (MacQueen, 1967; Jain, 2010; Chauhan et al., 2016b). Through an iterative approach, the K-means algorithm computes the Euclidean distance between the data points (pixel value) to its nearest centroid (cluster). The iteration converges when the objective function, i.e. the mean square root error of Euclidean distance, reaches the minimum. This is when each of the pixels in the dataset is assigned to its nearest centroid (cluster). However, the K-means algorithm has the tendency to converge at local minima without reaching the global minimum of the objective function. Therefore, it is recommended to repeatedly run the algorithm to increase the likelihood that the global minimum of the objective function will be identified. The performance of the K-means algorithm is influenced predominantly by the choice of the cluster centres (Chauhan et al., 2016b).

The Fuzzy C-means (FCM) clustering procedure involves minimizing the objective function (Dunn, 1973)

$$J_{fcm}(Z; U; V) = \sum_{j=1}^{n} \sum_{i=1}^{k} (\mu_{ij})^m \left\| x_i^{(i)} - c_k \right\|^2 \quad (10)$$

where $c_k = \sum_{j=1}^{n} u_{ij} x_i$

$c_k$ is the $k^{th}$ fuzzy cluster centre, $m$ is the fuzziness parameter, $m.u_{ij}$ is the membership function.

Unlike k-means, FCM performs a sort of soft clustering, in the FCM iterative scheme each data point can be a member of multiple clusters (Dunn, 1973; Bezdek et al., 1987; Jain et al., 1999; Jain, 2010). This notion of Fuzzy clustering can be

controlled by using a membership function (Zadeh, 1965). The membership value ranges between [0,1] and by selecting different membership values, the distance function can be regularized "loosely" or "tightly" and certain material phases with low volume fraction can be conserved from being clustered in adjoin cluster boundaries. However, it is essential to test different combination of membership values with several centroid centres (segmentation classes) to obtain reliable results.

### 2.2.2. Supervised Machine Learning Technique

Similar to unsupervised techniques, the objective of the supervised machine learning technique is to separate data. The advantage supervised technique offers compared to unsupervised technique is that it is effective in separating non-linear separable data (Haykin, 1995; Bishop, 2006). Datasets can be linearly separable if the points in the dataset can be partitioned into two classes using a threshold function (threshold should not be a piecewise discontinuous function). Loosely speaking the

10 threshold function fits a line to produce the partition. On the contrary, if we try to fit a threshold function to a substantially overlapped dataset─ this usually leads to wrong partitioning (Bishop, 2006; Haykin, 1995). Therefore, such dataset is regarded as linearly in-separable alias non-linear separable dataset (Bishop, 2006). In a supervised technique, the prediction is made by a model. The model is a mathematical function, which fit a line or a plane in between linearly or non-linearly separable data to classify them into different categories. The model's ability or intuition, where to place the line or plane in between the

15 dataset to clearly separate (classify) them─ is based on its (model) apriori knowledge of the dataset─ this apriori knowledge is called the training dataset. Therefore, unlike unsupervised technique, the supervised model needs to be trained on a subset of the dataset. The training dataset is the only 'window' through which, the model knows some pattern about the linear or non-linear separable dataset. How good, the model has acquired the knowledge of the training dataset, determines its success in prediction. If it has learned the training data accurately, it picks up noise along with the pattern and loses its ability of

20 generalization, thus fails when introducing to (unknown) separable-dataset. Alternatively, if the model has 'vaguely' learnt the training dataset; could be due very less training dataset, or if the model is too simple or complex to learn. This could also lead to failure in prediction. Therefore, to manage a good tradeoff, cross-validation techniques are used to monitor the learning rates of the model (Haykin, 1995).

Support vector machine (SVM) (Haykin, 1995) and its modified version least square support vector machine (LSSVM) are one such category of supervised ML technique (Suykens and Vandewalle, 1999) and use the principles mentioned above. The plane separating the data is termed as a hyperplane. The hyperplane has a boundary around it, which is called the *margin* and the data points that lie closest or on the margin are called the *support vectors*. The width of the margin governs the tradeoff, i.e. if the model is overfitted or under fitted to the training dataset; and can be verified through cross-validation techniques. If

the width of the margin is two narrow (high learning rate), the model is overfitted (high variance) to the training dataset and will lose it generalization capability and may not separate the linear or non-linear separable (unknown) data accurately. If the width of the margin is too wide (very low learning rate), the model is under fitted (high bias) to the training dataset and will

fail. Optimal learning model has a just the appropriate width, to maintain the generalization and also learn the patterns in the dataset.

If the training dataset is non-linear and inseparable in a 2D coordinate system, it is useful to project the dataset in 3D coordinate system− thus, by doing so, the added dimension (3D) helps to visualize the data and find a place fit a hyperplane to separate them (T. M. Cover, 1965). So, SVM and LSSVM use the principle of cover theorem (T. M. Cover, 1965) to project the data into a higher dimension to make them linearly separable and transform them back the original coordinate system (Suykens and Vandewalle, 1999). Hence what type of projection is to be performed by the SVM or LSSVM is done by choosing appropriate kernel function (van Gestel et al., 2004). This gives them the capability to attain the knowledge of the data and also preserve
the generalization behaviour of the model or the classifier. In the original or the 2D coordinate system, the hyperplane, is no longer a line but a convex-shaped curved which has clearly, separated the data and suitable margins to the support vectors. Here, 3D implies a 2+1 dimensional space which consists of two spatial dimensions that correspond to the coordinated of the pixels position in the image and the third dimension to that of the greyscales that evolve as a result of the LSSVM machine learning.

### 2.2.3. Ensemble Classifiers Technique

As the name implies ensemble classifier is an approach, where the decision of several simple models is considered to improve the prediction performance. The idea behind using ensemble methods emulates from a typical human approach of exploring several options before making a decision. The ensemble technique is faster compared to supervised techniques. Basically, the
20 evaluation of the decisions predicted by the simple models can be either done sequentially (Bragging or Boosting) or in parallel (Random Forest). Our toolbox used the sequential approach with a variation Bragging and Boosting for classification. These Bragging and Boosting used tree learners (Seiffert et al., 2008; Breiman, 1996), inherited from the MATLAB® libraries.
The main difference between Bragging and Boosting are as follows. Bragging generates a set of simple models, first trains these models with the random sample and evaluates the classification performance of each model using the test subset of data.
In the second step, only those models whose classification performance was low are retrained. The final predictive performance rate of the Bragging classifier− is an average of individual model performances. This approach minimizes the variance in the prediction−, meaning if several Bragging classifiers are generated from same sample of data−, their prediction capability when exposed to the unknown dataset, will not differ much. The main difference of Boosting to Bragging is that; Bragging retrain selected models (high misclassification rate) with the complete training dataset until their respective accuracy increases.
Whereas, in Boosting the size of the data which has been misclassified in increased in ratio to the data which has been accurately classified− and thereafter *all* the models are retrained sequentially. The predictive performance is calculated same as in Bragging by averaging the predictive performance rate of the individual models. This approach of Boosting minimizes the bias in the prediction.

## 2.3. Performance

It is necessary to monitor the performance of an ML model. This ensures that the trained modelled does not overfit or underfit with the training dataset. The main reason for overfitting and underfitting of the model with the training dataset is directly proportional to the complexity of the ML models. However, the consequence is that an overfitted trained ML model will capture noise along with the information pattern from the training dataset and will lose its ability of generalizability. Hence leading to inaccurate classification when exposed to the unknown dataset; as it has high variance toward the training dataset. On the opposite side, when the ML model underfit with the training dataset, it is unable to learn or capture the essence of the training dataset; this can happen either due to a choice of a simple type model (E,g linear instead of quadratic) or very less amount of data to build a reliable model. As a consequence the ML fails to predict as it has low variance towards the training dataset (Dietterich, 1998). So, the performance of the ML model; (low variance and low bias) is an indication of how accurate it can predict. The above explanation is valid for supervised ML techniques. For unsupervised clustering techniques where there isn't any model available to train. The quality of the classification is judge from the classified result. One such commonly used metrics is the Entropy (Stehl 2002; Meila 2003 and Amigó et al., 2009). In CobWeb, the performance of the ML models and the quality of the classification can be evaluated using 10-fold Cross-Validation, Entropy, Receiver Operational Characteristics (ROC). The explanation of these methods is briefly described in the subsection below. For detailed information the readers are referred to Stehl (2002) (Dietterich, 1998; Bradley, 1997) and references within.

### 2.3.1. 10-K fold Cross-Validation

The idea for K-fold cross-validation was first recommended by Larson (1931). K-fold cross-validation is a performance evaluation technique which checks the overfitting and underfitting of the ML model. In the K-fold technique, the training data is divided into k partitions. Thereafter, the ML model is trained with k-1 partition of data and tested on a withheld $k^{th}$ subset of data that has not been used for training. This process is repeated k-times, through this each data point in the training dataset get to be tested at least once, and is used, for training k-1 times. As it can be seen, this approach should significantly reduce the overfitting (low variance) as most of the data is used for testing and underfitting (low bias) as almost all the data is used for training. As from empirical evidence in k = 10 is preferred.

### 2.3.2. Entropy

The Entropy of a class reflects how the members of the *k* pixels are distributed within each class; the global quality measure is by averaging the entropy of all classes.

$$\text{Entropy} = - \sum_j \frac{n_j}{n} \sum_i P(i,j) \times \log_2 P(i,j) \qquad (11)$$

Where $P(i, j)$ is the probability of finding an item from the category $i$ in the class $j$, where $n_j$ is the number of items in class $j$ and $n$ the total number of items in the distribution.

### 2.3.3. Receiver Operational Characteristics

Receiver Operational Characteristics (ROC) curves are one of the popular methods to cross-validation of ML model performance (probability of models correct response *P(C)* to the predicted result*) (Bradley, 1997). It has three variables:

$$\text{Accuracy (1 - Error)} = \frac{T_p + T_n}{C_p + C_n} = P(C) \tag{12}$$

$$\text{Sensitivity (1 - } \beta) = \frac{T_p}{C_p} = P(T_p) \tag{13}$$

$$\text{Specificity (1 - } \alpha) = \frac{T_n}{C_n} = P(n) \tag{14}$$

Where, $T_p$ $and$ $T_n$ are the true positive and true negative examples and $C_p$ $and$ $C_n$ are the total number of true positive and true negative examples obtained from the training dataset.

Probability of false positive is $P(F_p) = \alpha$

Probability of true positive is $P(T_p) = (1 - \beta)$

The accuracy is determined by calculation area under the curve (AUC), and the simplest way to do this was by using trapezoidal approximation.

$$\text{AUC} = \sum_i \left\{ (1 - \beta_i \cdot \Delta\alpha) + \frac{1}{2}(\Delta(1 - \beta). \ \Delta\alpha) \right\} \tag{15}$$

## 3. Toolbox and functionalities − CobWeb Key Features

The first version of CobWeb offers the possibility to read and to process reconstructed XCT files in both .tiff and .raw formats. The graphical user interface (GUI) is embedded with visual inspection tools to zoom in/out, cropping, colour, and scale, to assist in the visualization and interpretation of 2D and 3D stack data. Noise filters such as non-local means, anisotropic diffusion, median and contrast adjustments are implemented to increase the signal-to-noise ratio. The user has a choice of five different segmentation algorithms, namely K-means, Fuzzy C-means (unsupervised), least square support vector machine (LSSVM) (supervised), bragging and boosting (ensemble classifiers) for accurate automatic segmentation and cross-validation. Relevant material properties like relative porosities, pore size distribution trends, volume fraction (3D pore, matrix, mineral phases) can be quantified and visualized as graphical output. The data can be exported to different file formats such as Microsoft® Excel (.xlsx), MATLAB® (.mat), ParaView (.vkt) and DSI studio (.fib). The current version is supported for Micosoft® Windows PC operating systems (Windows 7 and 10).

The main GUI window panel divides into three main parts (Figure 2), the tool menu strip, the inspector panel, and the visualization panel. The tool strip contains menus for zoom in and out, pan, rotate, point selection, colour bar, legend bar, and

measurement scale functionalities. The inspector panel is divided into subpanels where the user can configure the initial process settings such as segmentation schemes (supervised, unsupervised, ensemble classifiers), filters (contrast, *non-local means*, *anisotropic* filter, *fspecial*), and distance functions (link distance, Manhattan distance, box distance) to assist segmentation and geometrical parameter selection for image analysis (REV, porosity, pore size distribution (PSD), volume fraction). The display

subpanel *records*, displays the 2D video of the XCT stack and the respective histogram. History subpanel is a *uilistbox* that displays errors, processing time/status, processing instruction, files generated/exported and executed callbacks. Control subpanel is an assemblage of *uibuttons* to initialize the XCT data analysis process and the progress bar. Visualisation panel is where the results are displayed in several resized windows, which can be moved, saved or deleted. The pan-windows displayed inside the visualization module are embedded with *uimenu* and *submenu* to export, plot and calculate different variables like

porosity, PSD, volume fraction, entropy, or receiver operational characteristics. To get the desired user functionalities, MATLAB® internal user-interface libraries were inadequate. Therefore, numerous specific adaptions are adopted from Yair Altman's undocumented MATLAB® website and the MATLAB® File Exchange community. Specifically, the GUI Layout Toolbox of David Sampson is used to configure the CobWeb GUI layout; the preprocessing *uitable*, uses the MATLAB® java-component; it was designed using *uitable* customization report provided by (Altman, 2014).

As a stand-alone, the CobWeb GUI can be executed on different PC and HPC clusters without any license issues. The framework of CobWeb 1.0 is schematically illustrated in Figure 3 and the direction for the arrow (left to right) represents the series in which the various functions are executed. The backend architecture can be broadly classified into three different categories, namely:

- Control module

- Analysis module

- Visualization module

### 3.1.1. Control module

Initially, the main figure panel is generated, followed by the tool strip dividing the main figure into different panels and subpanels as shown in Figure 2. After that, the control buttons *Load, Start, Stop, Volume Rendering* and *Clear* are created,

initialized and the relevant information is appended in the main structure. Ideally, at this point, any button can be triggered or activated. However, on doing so, an exception will be displayed in the history subpanel, indicating the next arbitrary steps. That is, to first load the data by pressing the *Load* button, where the *Load* function checks the file properties, loads the data in .tiff and .raw format, creates and displays 2D video of the selected stack, save the video file in the current folder, and updates the respective variables to the main structure. The *Stop* button (*Stop* function) ends the execution. However, when the

processing is inside a loop, the *Stop* function can break the loop only after the i^th iteration. The *Clear* button (*Clear* function), deletes the data and clears all the variables in the main structure, resetting the graphical window.

### 3.1.2. Analysis module

The next step is data processing; triggered by pressing the *Start* button, which activates the *Start* function. The *Start* function concatenates the entire analysis procedure and is shown as Start () in Figure 3. is a function of densely nested loops, the bullet points and the sub-bullet points shown in Figure 3, symbolizes the outer and the inner nested loops. Initially, the data is gathered, and a sanity check is performed to evaluate, if the user, selected the relevant checkboxes and respective suboptions in the preprocessing *uitable*. If the checkboxes are not selected, an exception alert is displayed in the *History* panel, highlighting the error and suggesting the next possible action. The next loop is the image modification loop, where the user inputs are required. These inputs are desired classes for segmentation, the image resolution, and the representative slice number. Thereafter, the representative slice is displayed on a resizable pan-window inside the visualization panel shown in Figure 2. Further, an option to select a region of interest (ROI) is proposed, which can be accepted or rejected. If accepted, a REV is cropped from the 3D image stack based on user-defined ROI dimensions. On rejection, the complete 3D stack is prepared for processing.

The next step is the segmentation process; an unsupervised or supervised algorithm is initialized based on the selection made by the user in the preprocessing *uitable*. Hereafter, the programming logic implemented at the data access layer (also known as back-end) for unsupervised and supervised segmentation schemes is briefly explained. It is an easy, one-step process in the case of unsupervised techniques i.e. based on the options selected in the preprocessing *uitable*, the image is filtered and subsequently, segmented. But, for unsupervised segmentation technique, Fuzzy C-means (FCM), additional user input is required. A positive decimal number $x$, where $x$ is equal to $1 \le x \ge 2$ to set the membership criteria; when pixel values of different phases (ex. Rotligend sandstone, Grosmont carbonate rock etc.) are in close vicinity or subsets of each other, FCM uses, the membership criteria to constrain the segmentation 'loosely' or 'tightly' with the purpose to segregate different phases (Chauhan et al., 2016b). In the case of supervised segmentation schemes (LSSVM, Bragging and Boosting) apriori information, also known as feature vector dataset or training dataset, is required to train the model(s) (Chauhan et al., 2016a; Chauhan et al., 2016b), and consequently, the trained model is ready to classify the rest of the dataset. The following five steps accomplish this procedure:

- First, the visualization panel displays a single 2D slice of the REV or 3D image stack in a resizable pan-window. The embedded *uimenu* in the pan-window offers to use the *subuimenu* options to feature vector selection, training and testing.
- Second, by pressing the *subuimenu* option *Pixel Selection,* the feature vector selection (FV) performs. The *Pixel Selection* callback function initializes the subroutine *uPixelSel(),* which sequentially displays a *uitable* in a resizeable pan-window. The *uitable* contains columns *Features*, *X-Coordinate*, and *Y-Coordinate*, which is, for example, the pixel coordinates of the pore, matrix, minerals, noise/speaks. This is a mandatory step, to build the training dataset. The user enters this information in the respective columns of the *uitable*.

- In the third step, the user has to identify features, such as pores, minerals, matrix, noise/specks, in the 2D image, using zoom in and out tools available in the toolbar. The X-coordinates and Y-coordinates of the identified features need to be extracted using the data cursor tool, also available in the toolbar. If satisfied, the user can enter, the features and the corresponding X,Y coordinates in the *Pixel Selection uitable*.
- In the fourth step, the data is gathered and exported for training. This is done by pressing the export button placed on the *uitable* pan-window; which initiates the subroutine *uExportTable( )*. The export subroutine collects a total of 36 (6 x 6) pixel values in the perimeter of the user-specified X, Y coordinates in the *uitable*.
- In the fifth step, the model is trained. This is done by using the *subuimenu* in the 2D pan-window. As and when the *training* is finished, a notification appears on the *History* panel. Thereafter by pressing the *testing* option in the *subuimenu* the complete REV or 3D stack can be segmented.

A progress bar offers to monitor the state of the process. Further, the *History* window displays information related to processing time, implemented image filters and the segmentation scheme. Finally, all relevant information and the segmented data is appended to the main structure.

### 3.1.3. Visualisation Module

Once the processing is finished, the segmented data can be visualized in the 2D format using *Plot* button or in a 3D rendered stack using *VolRender* button. Figure 3 depicts the nested loop structure of the *Plot ( )* and *VolRender ( )* callback functions. On initialization, the *Plot( )* callback accesses the main structure, and plots the segmented 2D image of the segmented slice consecutive, in a resizable pan-window in the visualization panel. The displayed pan-window is embedded with a *uimenu* and corresponding *subuimenu*. The *uimenu* items and the *subuimenu* options are

- Geometrical Parameters → Porosity, Pore Size Distribution, Volume Fraction
- Performance → Entropy, Receiver Operational Characteristics (ROC), 10-fold Cross Validation
- Export Stack → ParaView, Raw

The methods used to calculate geometrical parameters and validation schemes are benchmarked in (Chauhan et al., 2016a) (Chauhan et al., 2016b). Therefore, the selection of desired options initialize respective subroutines (uPoreSzVol, uCalVal, uExport) and plot the results as shown in Figure 2. If required, the export of these parameters (Porosity, PSD, Volume Fraction, Entropy, ROC; 10-fold Cross Validation) is possible to Excel, ASCII or MATLAB$^{®}$ for further statistical analysis. Using the *Export Stack* item, the export of the 3D segmented volume to ParaView (.vtk files) or as .raw format files is feasible for the purpose of visualization or DRP analysis. The volume rendering functionalities of CobWeb 1.0 is simple in comparison to ParaView or DSI studio. The *VolRender ( )* function renders the 3D data set using orthogonal plane 2-D texture mapping technique (Heckbert, 1986) and is best suited for OpenGL hardware. The user has the option to render the 3D stack in the

original resolution or at lower resolution; the lower resolution enhances the plotting speed but degrades the image quality by 10-folds. Due to this, we recommend to export the 3D stack to ParaView or DSI studio for visualization. This concludes the description of the section toolbox and functionalities. For more information on the usage of the graphical user interface the user manual can be consulted, which is available as supporting information.

In the following, sections the CobWeb toolbox is demonstrated by means of three showcase examples, which are briefly introduced in terms of underlying imaging settings, research question and challenges for image processing.

## 4. Materials and Methods

### 4.1. Gas-Hydrate bearing Sediment

The in-situ synchrotron-based tomography experiment and post-processing of synchrotron data conducted to resolve the microstructure of gas hydrate-bearing (GH) sediments are given in detail by Chaouachi et al. (2015), Falenty et al. (2015), and Sell et al. (2016). In brief, the tomographic scans were acquired with a monochromatic X-ray beam energy of 21.9 KeV at Swiss Light Source (SLS) synchrotron facility (Paul-Scherrer-Institute, Villigen, Switzerland) using the TOMCAT beamline (Tomographic Microscope and Coherent Radiology Experiment; Stampanoni et al. 2006). Each tomogram was reconstructed

from sinograms by using the gridded Fourier transformation algorithm (Marone und Stampanoni 2012). Later, a 3D stack of 2560 x 2560 x 2160 voxels (volume pixels) was generated resulting in a voxel resolution of 0.74 μm and 0.38 μm at 10-fold and 20-fold optical magnification.

#### 4.1.1. Dual Filtering of Gas-Hydrate bearing Sediment

The ED artefact is the high and low image contrast seen, between the edges, of the void, quartz and GH phases, in the GH

tomograms. It certainly, aids in clear visual distinction, of these phases, but, become a nuisance during the segmentation process. Several approaches to reduce ED artefact in GH tomograms and its effect on segmentation and numerical simulation have been discussed in (Sell et al., 2016). Based on our experience, a combination of the non-local means (NLM) filter and the anisotropic diffusion filter (AD), implemented using Avizo (ThermoFisher Scientific), works best in removing ED artefacts for our GH data. In short, AD was used for edge preservation and NLM for denoising. In this study, the NLM filter was set to

a search window of 21, local neighbourhood of 6 and a similarity value of 0.71. The NLM filter was implemented in 3D mode to attain desired spatial and temporal accuracy and was processed on a CPU device.

#### 4.1.2. Gas Hydrate (GH) bearing Sediment Dual Clustering

The edge enhancement effect was significant in all the reconstructed slices of the GH dataset. The ED effect was noticeable around the quartz grains, with high and low pixel intensities adjacent to each other. The high-intensity pixel values (EDH)

were very close to GH pixel values, while the low-intensity pixel values (EDL) showed a variance between noise and void

phase pixel values. Therefore, immediate segmentation performed on the pre-filtered GH datasets using CobWeb 1.0 resulted in misclassification. Further parameterizing and tuning the unsupervised (K-means) and supervised (LSSVM) modules of CobWeb 1.0 specifically, distance function (i.e., functions euclidean distance *sqeuclidean*, sum of absolute differences *cityblock*, and *mandist*) and different permutation and combination between of kernel type, bandwidth and cross-validation parameters, showed significant improvement, but the segmentation was still not optimal. The aim was to eliminate the ED features completely without altering the phase distribution between GH and the void. This prompted to develop a GH-specific workflow as explained below. The appendix provides the MATLAB® script for this workflow comprised of 5 steps:

**Step 1: Filtering and REV selection**

Four REVs of size $4 \times 700^3$ were cropped from the raw (16 bit) data stack. These REVs were dual-filtered using AD and NLM filters (see section 4.1.1). Figure 5 depicts a 2D dual-filtered image from REV1. In this study, the NLM filter was set to a search window of 21, local neighbourhood of 6 and a similarity value of 0.71. The NLM filter was implemented in 3D mode to attain desired spatial and temporal accuracy and was processed on a CPU device.

**Step 2: K-means clustering**

After, dual-filtration (Step 1), it was essential to segregate the noise, edge enhancement effects and different phases into labels of various classes. This was accomplished by K-means segmentation. In order to capture all the phases accurately along with noise and ED affects a segmentation process with up to twenty class labels was needed and performed. As a result class seven captured all the desired phases (noise, edge enhancement low intensities (EDL), void, quartz, edge enhancement high intensities (EDH), GH).

**Step 3: Indexing**

In the next step, the purpose was to retrieve pixel values of various phases from the dual-filtered REV stacks. The indexing scheme is the following:

- First, through visual inspection of the segmented image (step 2) different phases and their corresponding labels where identified, shown in Table 1.
- Thereafter, pixel indices of these phases, where extracted from the segmented image based on their labels.
- Further, these indices were used as a reference mask to retrieve pixel values of the phases from the 16-bit raw REV stacks.

The obtained pixel values represent noise, void (liquid), EDL, quartz, EDH, and GH phases in the raw images. Then, the histogram distribution of the pixel values in each phase was plotted. The skewness of the histograms was investigated where the max, min, the mean and standard deviation for each of the histogram was calculated. Thereafter, max and min of the histograms where compared, and the indexing limits were adjusted, for as-long-as there was no overlap found amidst the histogram boundaries.

**Step 4: Rescaling raw REV**

In this step, the raw pixel values of the respective phases, i.e void, quartz, and GH, were replaced by their mean values, with an exception for EDH pixel values. The latter (EDH pixels) were replaced with the mean value of quartz. These assignments lead to optimal segregation of the phase boundaries in the raw dataset and finally to the elimination of the ED effect.

**Step 5: K-means clustering**

Finally, the re-scaled raw REV was segmented into three class labels using K-means segmentation to obtain the final result.

## 4.2. Grosmont Carbonate Rock

The digital rock images of the Grosmont carbonate rock were obtained from the FTP server GitHub (http://github.com/cageo/Krzikalla-2012) used in the benchmark study published by (Andrä et al., 2013a, 2013b). The Grosmont carbonate rock was acquired from Grosmont formation Alberta, Canada. The Grosmont formation was deposited during upper Devonian and is divided into four facies members, LG UG-1, UG-2, and UG-3 (bottom to top). The sample was taken from UG-2 facies and is mostly composed of dolomite and karst breccia (Machel and Hunter, 1994; Buschkuehle et al., 2007). Laboratory measurements of porosity and permeability reported by (Andrä et al., 2013b) are around 21 % ($\phi = 0.21$) and $\kappa = 150$ mD $-$ 470 mD, respectively. The Grosmont carbonate dataset was measured at the high-resolution X-ray computed tomographic facility of the University of Texas with an Xradia MicroXCT-400 instruments (ZEISS, Jena, Germany). The measurement was performed using 4x objective lenses, 70 kV polychromatic X-ray beam energy, and a 25 mm CCD detector. The tomographic images were reconstructed from the sinograms using proprietary software and corrected for the beam hardening effect, which is typical for lab-based polychromatic cone-beam X-ray instruments (Jovanović et al., 2013). The retrieved image volume was cropped to a dimension of $1024^3$ with a voxel size of 2.02 μm.

## 4.3. Berea Sandstone Rock

The Berea sandstone digital rock images were part of a benchmark project published by Andrä et al. (2013a, 2013b) and obtained from the *GitHub* FTP server. The Berea sandstone sample plug was acquired from Berea Sandstone Petroleum Cores [TM] (Ohio USA). The porosity value of 20 % ($\phi = 0.20$) was obtained using a Helium pycnometer AccuPyc[TM] 1330 (Micromeritics Instrument Corp., Germany) and a Pascal-Mercury porosimeter (Thermo Scientific [TM] ) as described in Giesche (2006). The permeability ranges between $\kappa = 200$ mD and $\kappa = 500$ mD as reported by Andrä et al. (2013b). Machel and Hunter (1994) identified minerals using a polarized optical microscope and a scanning electron microscope and reported a mineral composition of Ankerite, Zircon, K-feldspar, Quartz, and Clay in the Berea sandstone

sample. The synchrotron tomographic scans of Berea sandstone were also obtained at the SLS TOMCAT beamline. The beam energy was monochromatized to 26 keV for optimal contrast with an exposure time of 500 ms. This resulted in a 3D tomographic stack with a dimension of $1024^3$ voxels with a voxel size of 0.74 μm.

## 5. Result and Discussions

### 5.1. Data Selection

The represented elementary volume (REV) selection basically, was a combination of visual inspection and consecutively segmenting and plotting trends in relative porosity, pore size distribution and volume fraction. This was done by loading the complete stack in the CobWeb software, during the loading process a 2D movie of the tomogram is displayed in the display window and saved in the root folder. Carefully monitoring the movie gives an objective evaluation of the heterogeneity of the respective XCT sample. We observed, several sub-sample volumes at various location (X, Y) and depth (Z) inside the XCT tomograms. Thereafter, based on a subjective visual consensus different ROIs where selected, cropped, segmented and their respective geometrical parameter where intercompared. The main indicator, however, was the porosity trend; i.e when regression coefficient R2 value was close to zero, it was an indicator that its sub-volume has accumulated the heterogeneity along the z-axis of the sample. Therefore, based on the trend analysis approach, the sub-volume dimension, were the R2 value was close to zero was chosen as the suitable REV.

In the case of Berea sandstone, four different ROIs were investigated; whereas Grosmont carbonate rock seven different ROIs where need to identify the best REVs. Cubical stack size in between $300^3$ to $700^3$ slices were tested and later established that stack size around $480^3$ suited the best. Through our previous scientific studies on the GH sediments (Sell et al., 2016; Sell et al., 2018) we were aware of the best-suited REVs and established that stack size of $700^3$ was an appropriate stack size. The identification of best REV for Grosmout was relatively tedious compared to Berea sandstone and GH sediment; due to the low resolution and microporosity present in the Grosmont tomograms. Figure 4 shows the chosen ROIs of Berea, Grossmont and GH dataset and Figure 6 and Figure 8 show the surface plot for respective REVs.

### 5.2. Data Processing

In the case of Berea sandstone, the 3D reconstructed raw images ($1024^3$) had sufficient high resolution and contrast, thus did not show any noticeable change to the filtration. Whereas, the XCT images ($1024^3$) of the Grosmont carbonate rock needed a non-local means filtering which yielded in better visualization and performance results compared to those enhanced with anisotropic diffusion filter. However, for GH synchrotron dataset, the CobWeb 1.0 filters were insufficient to normalize the edge enhancement artefact, several attempts were made to remove the edge enhancement effect using single filters and in combination with supervised techniques. But they did not yield desirable results. The edge enhancement artefact pixels values wherein very close vicinity to the GH sediment pixels. Therefore, preprocessing with single filters despite using appropriate

settings could not normalize enhancement artefact to a reasonable range of. Despite tailoring customized training dataset using a representative slice− due to large standard deviation in the edge enhancement artefact values, GH was systematically misclassified as ED as the pixel values deviated away from the trained model. An alternative approach was to create different training dataset using several representative slices and introduce the unknown stack of data for classification in batches of 100 5   slices. This regularization trick for us did not represent a good norm for supervised ML classification.

Hence, through the experience gained in (Sell et al., 2016) for us dual-filtration was one of the best approaches we could include in the preprocessing step. This dual-filtering did not remove the ED completely rather normalize it to a reasonable range. Through the approach of rescaling and (hard) K-means segmentation (dual-segmentation) we were absolutely sure that 10   the ED artefact has been removed. This dual filtering scheme is explained in section 4.1.2. It is to be noted that, the NLM filter is hard-coded as 2D in the CobWeb standalone version (GUI). But, by tweaking or modifying the source code we could initially pre-process the XCT images using NLM 3D filtration and thereafter subjected it to segmentation.

In general, our observation is that, depending on the resolution of the dataset, the fixed parameters of NLM and other filters 15   should do a fairly good job. In case, there still exists noise and artefacts we recommended that the supervised techniques be used.  The supervised techniques offer the possibility to select the residual noise or artefact pixel values before or after the filtration (pre-processing) through proper feature vector selection, and further training the appropriate model and performing classification. Through which, the existing noise and artefact can be isolated and segmented as separate labels. Another alternative option could be to pre-process the data with desired filters data and imported the data into CobWeb for segmentation 20   and analysis.

Another issue has to be explained in more detail in the implementation of the image segmentation. CobWeb 1.0 uses a slice-by-slice 2D approach. It was observed that the ML techniques tend to underestimate porosity values compared to manually segmented analysis at a REV scale size $> 500^3$. This substantial degree of uncertainty is caused due to 2D slice-by-slice 25   processing rather than the ML techniques. The 2D slice-by-slice approach, passes only, the spatial information (X and Y coordinate direction) to the ML algorithms, and the ML algorithm ends up sorting the intensity variation in the spatial domain (local optimum). Therefore, the lack of spatial-information (Z coordinate direction) restricts the degree of freedom to find, a global optimum. In other words, changes, due to bedding (sedimentary rock) or microporosity (carbonate rocks) in the rock texture, are represented as a sudden spike or dip in porosity values; which appear as artefact or anomalies− and are often 30   discarded. We acknowledge this issue and correction will be implemented in the future software version; in the current workflow it has not been accounted for (CobWeb 1.0). The 2D slice-by-slice processing scheme is much faster compared to the 3D approach. So, the choice of 2D processing for this research study was made to make it affordable to compute on desktop, laptop for near real-time and onsite evaluation.  The inaccuracies in porosities are compensated by calculation of the mean porosity of the complete stack.

### 5.3. Multiphase Image Segmentation

The major problem for all multiphase segmentation is that phase having intermediate greyscale values gets sandwiched between two different phases. These intermediate phases sometimes represent some of the vital material property such as connectivity. Therefore, it is vital to emphasis, how ML can assist in issues related to multiphase segmentation. In a practical sense, machine learning tries to separate grayscale values into disjoint sets. The creation of these disjoint sets is commonly done in two ways

1) By binning the greyscale values to the nearest representative values which is iteratively updated using an optimization function. This optimization function can be a simple regression or distance function (Jain et al., 1999), commonly used in unsupervised techniques.

2) Second, by regularizing pre-trained models which store certain pattern information of the datasets such as topology features, contour intensities, pixel value etc. (Hopfield, 1982; Haykin, 1995; Suykens and Vandewalle, 1999). Or by using a voting system in a bootstrap ensemble of linear models (Breiman, 1996).

So, in this processes, the intermediate greyscale values corresponding to low volume fraction which shows multi-modal distributions are merged with greyscale values of high volume fraction to create disjoint boundaries. Through which the intermediate phase information is misclassified and hence destroyed. One way to overcome this problem is by using, supervised techniques such as LSSVM or Ensemble classifiers. When constructing a training dataset (feature vector selection), careful selection of intermediate phases as a sufficiently large sample size compared to the predominant phases will preserve the intermediate phases. And, the likelihood that the trained model will identify them and cluster them separately is higher (Chauhan et al., 2016a). In this study, in particular, we made tests using supervised techniques (LSSVM, Ensemble classifies) and unsupervised technique (FCM) but the results weren't superior compared to K-means. Therefore, we choose K-means as it was faster compared to other ML techniques. Since we have used K-means for segmentation, it is necessary that we justify the performance of K-means in terms of accuracy and speed. In the current research work, since we have used unsupervised technique, it safe to say that, accuracy and speed are directly proportional to starting point (initial location) in the segmentation process. Meaning, the closer the starting point (initial location) is to the global minima— faster will the algorithm converge and even so better is the performance (accuracy & speed). But, in unsupervised technique by default the choice of the starting point is through random seed unless explicitly specified. So, in the case of the dual segmentation approach used for segmentation, the intuition was to capture all the material phases, including the edge enhancement artefact, speck and noise etc. in the first step and thereafter in the second step to rescale them to the plausible phases. Hence, in the first step the 20 clusters were initialized using random seed. And, after the rescaling processes, we were aware of the initial locations which we used as a starting point (initial location) to assist the algorithm to move towards identifying correct phases. Therefore, we could increase both the speed and accuracy of K-means. In the case of Berea sandstone, the segmentation was restricted to

four clusters out of which three phases can be clearly seen. The first two phases being pores and rock, in the third phase minerals Ankerite, Zircon, K-feldspar, Quartz, and Clay have been classified into single mono-mineral phase and the fourth phase comprise of small scale features like residual speck and noise pixels. The Grosmont carbonate sample was also segmented into four clusters, comprising of pore, pore inclusions, Calcite and brightness inhomogeneities of noise classified and the fourth phase.

Note that the emphasis of this study was to demonstrate the capabilities of CobWeb and removal of edge enhancement segmentation, through dual filtration and dual segmentation schemes. Detailed verification with LSSVM and ensemble classifiers falls therefore outside the scope of this work and readers are referred to the previous work from (Chauhan et al., 2016a) based on which CobWeb is developed. That work benchmarks different ML algorithms and quantifies their respective accuracies and performances.

## 5.4. Estimation of Relative Porosity and Pore Size Distribution

The pore size distribution (PSD) of the respective REVs were calculated using the CobWeb PSD module. The PSD module is based on an image processing morphological scheme (watershed transformation) suggested by Rabbani et al. (2014). As stated by Rabanni, in (Rabbani et al., 2014), the aim is to breakdown the monolithic void structure of rock into specific pores and throats connecting each other. (Rabbani et al., 2014) used unsegmented images and performed image filtration and thereafter segmented using watershed transformation. In our case, the tomograms were already pre-processed and segmented using ML techniques. These images are converted to binary images and thereafter subjected to the image processing distance function (Rosenfeld, 1969) and the watershed algorithm (Myers et al., 2007) to extract pores and throats. City-block distance function is used to locate the void pixels (pores) and watershed with the 8-connected neighbourhood was used to obtain the interconnectivity. Since the watershed algorithm is very sensitive to noise, despite the preprocessing and ML segmentation median filter was applied before subjecting to the watershed segmentation. Thereafter, the mean relative porosity value obtained for Berea sandstone is $\phi =17.3 \pm 2.6$ %, whereas for Grosmont carbonates the mean porosity value is lower ($\phi =10.5 \pm 2.3$ %) shown in Figure 6. Particularly, in the case of Grosmont, after segmentation the obtained porosity value $\phi =10.5 \pm 2.3$ % is extremely low compared to the laboratory measurement $\phi =21$ % published in (Andrä et al., 2013a). The exact reason is not known, but could also be partly attributed to sub-resolution pores which couldn't be captured do to low resolution obtained through XCT measurement. The regression coefficient value of $R^2 = 0.092$ for the Berea sandstone porosity trend indicates that porosity remains constant throughout the REV sizes chosen, and therefore consolidated for scale-independent heterogeneities. In the case of Grosmont carbonate rock, the chosen REV size was the best out of five obtained, which consolidate again for scale-independent heterogeneities. The average pore size distribution thus obtained was 6.70 μm $\pm$ 0.68 μm and 14.21 μm $\pm$ 0.66 μm for Berea and Grosmont plug samples, respectively.

Similarly, the porosity and PSD of the four GH REVs were analyzed using CobWeb 1.0 and is shown in Figure 7. The low $R^2$ values of the porosity trends justify that, these GH REVs are scale-independent, and are an accurate representation of a large-scale system and are best suited for digital rock analysis. However, there is a high variance compared with the mean PSD values. The exact reason is unknown but may be due to the drastic increase and decrease of the quartz grains which can be seen in Figure 5 or could be that PSD requires much larger REV compared to that used for Porosity analysis. The first and last 2D slices of ROI 1 in Figure 5 show either non-isotropic or isotropic distribution of quartz grains, which might have contributed to the respective high and low standard deviation, seen in the porosity distribution. Figure 8 shows the surface and volume-rendered plots of REV 1 and REV 2, due to the high accuracy of segmentation the quartz grain, brine and GH boundaries are clearly segregated, and ED effect eliminated.

## 6. Conclusions and Outlook

This paper introduces with CobWeb 1.0 a new visualization and image analysis toolkit dedicated to representative elementary volume analysis of digital rocks. CobWeb 1.0 is developed on the MATLAB® framework and can be used as MATLAB® plugin or as a standalone executable. It offers robust image segmentation schemes based on machine learning (ML) techniques (unsupervised and supervised), were the accuracy of the segmentation schemes can be determined and results can be compared. Dedicated image processing filters such as the non-local means, anisotropic diffusion, averaging and the contrast enhancement functions help to reduce artefacts and increase the signal-to-noise ratio. The petrophysical and geometrical properties such as porosity, pore size distribution and volume fractions can be computed quickly on a single representative 2D slice or on a complete 3D stack. This had been validated using synchrotron datasets of the Berea sandstone (at a spatial resolution of 0.74 µm), a gas hydrate-bearing sediment (0.76 µm) and a high resolution lab-based cone-beam tomography dataset of the Grosmont Carbonate rock (2.02 µm). The gas hydrate dataset, despite its nanoscale resolution, was hampered with strong edge enhancement artefacts. A combination of the dual filtering and dual clustering approach is proposed to completely eliminate the ED effect in the gas hydrate sediments, and the code is attached as an appendix. The REV studies performed on Berea sandstone, Grosmont carbonate rock and GH sediment using CobWeb1.0 shows relative porosity trends with very low linear regression values of 0.092, 0.1404, 0.0527 respectively. CobWeb1.0 ability to accurately segment data without compromise on the data quality at a reasonable speed makes it a favourable tool for REV analysis.

CobWeb1.0 is still somewhat limited regarding its volume rendering capabilities, which will be one of the features to improve in the next version. The volume rendering algorithms implemented in CobWeb 1.0 so far do not reach the capabilities offered by ParaView or DSI studio, which relies on the OpenGL marching cube scheme. At present, the densely nested loop structure appears to be the best choice for systematic processing. As an outlook, vectorization and indexing approaches (*bsxfun, repmat*) have to be checked in detail to improve on processing speed. MATLAB® —Java synchronization will be explored further to configure issues related to multi-threading and visualization (Java OpenGL). Furthermore, a module CrackNet (crack network) is planned to be implemented, which will explicitly tackle the segmentation of cracks and fissures in geomaterials using

machine learning techniques and a mesh generation plugin (stl format) for 3D printing. Pore network extraction and skeletonization schemes such as modified maximum ball algorithm (Arand and Hesser, 2017) and medial axis transformation (Katz and Pizer, 2003) will be considered such that the data can be exported to open-source pore networks modelling packages such as Finite-difference method Stokes solver (FDMSS) for 3D pore geometries and OpenPNM (Gerke et al., 2018;

Gostick, 2017; Gostick et al., 2016).

## Code availability / Data availability

With regards to the code availability, the MATLAB® code for removal of edge enhancement artifacts from the gas hydrate bearing sediment is attached as appendix. The CobWeb executable as well as the user manual and The gas-hydrate bearing

sediment, XCT datasets are available to public on Zenodo repository http://dx.doi.org/10.5281/zenodo.2390943.

The CobWeb executable requires a MATLAB® runtime compiler R2017b (9.3), which can be downloaded and installed from https://ch.mathworks.com/products/compiler/MATLAB®-runtime.html. The XCT dataset of Berea Sandstone and Grosmount Carbonate Rock can be obtained from *GitHub* FTP server (http://github.com/cageo/Krzikalla-2012). The gas-hydrate XCT datasets are not publicly available.

## Author contribution

Contributor Roles Taxonomy (CrediT) is used to specify author contribution. https://casrai.org/credit/.

Swarup Chauhan conceptualized, investigated and performed the study. Further, implemented the machine learning workflow and graphical user interface design. Additionally, Swarup Chauhan performed the formal analysis and developed a software

code for the removal of the edge enhancement artefact using the dual clustering approach. Further contributions of Swarup Chauhan: data curation of the CobWeb software, writing the software manual, figures and writing, reviewing and editing the manuscript.

Kathleen Sell conceptualized, investigated and performed a case study on gas hydrates. Further, performed a study on the removal of edge enhancement artefacts and phase segmentation of methane hydrate X-ray tomograms (XCT). Also, she did a

formal analysis by implementing dual filtration approach to reduce the edge enhancement artefacts. Kathleen Sell participated in discussions to validate phase segmentation using the dual segmentation approach and was involved in writing, reviewing and editing the manuscript

Wolfram Rühaak was involved in the project administration of the CobWeb activities, provided resources with respect to graphical user interface (GUI) and inputs on improving GUI functionalities.

Thorsten Wille was involved in funding acquisition and sponsoring the CobWeb project, under the framework of the SUGAR (Submarine Gashydrat Ressourcen) III project by the Germany Federal Ministry of Education and Research grant number: 03SX38IH. He was involved in project administration and provided feedback on GUI functionalities

Ingo Sass was involved in the concept and funding acquisition for the CobWeb project, under the framework of the SUGAR (Submarine Gashydrat Ressourcen) III project by the Germany Federal Ministry of Education and Research (grant number:

03SX38IH). He also provided supervision, project administration, resources and periodic review to improve GUI functionalities.

**Acknowledgements**

We thank Heiko Andrä and his team at Fraunhofer ITWM, Kaiserslautern, Germany, for providing us with the synchrotron tomography benchmark dataset of the Berea sandstone. We also thank Michael Kersten, Frieder Enzmann and his group at the Institute for Geoscience, Johannes-Gutenberg Universität Mainz, for providing high-resolution gas hydrate synchrotron-data. The acquisition of the gas hydrate synchrotron-data was funded by the German Science Foundation (DFG grant Ke 508/20 and Ku 920/18). This study was funded within the framework of the SUGAR (Submarine Gashydrat Ressourcen) III project by the Germany Federal Ministry of Education and Research (BMBF grant 03SX38IH). The sole responsibility of the paper lies with the authors.

We thank Dr. Kirill Gerke, two anonymous reviewers and the editor Dr. Thomas Poulet for their valuable comments and suggestions which significantly improved the manuscript.

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

Table 1: Class labels of different phases

| Labels | Phases |
| --- | --- |
| 0 | Noise |
| 1 and 3 | Void (liquid) |
| 2 | Edge enhancement low intensities (EDL) |
| 4 | Quartz |
| 5 | Edge enhancement high intensities (EDL) |
| 6 and 7 | Gas hydrate |

**Figures**

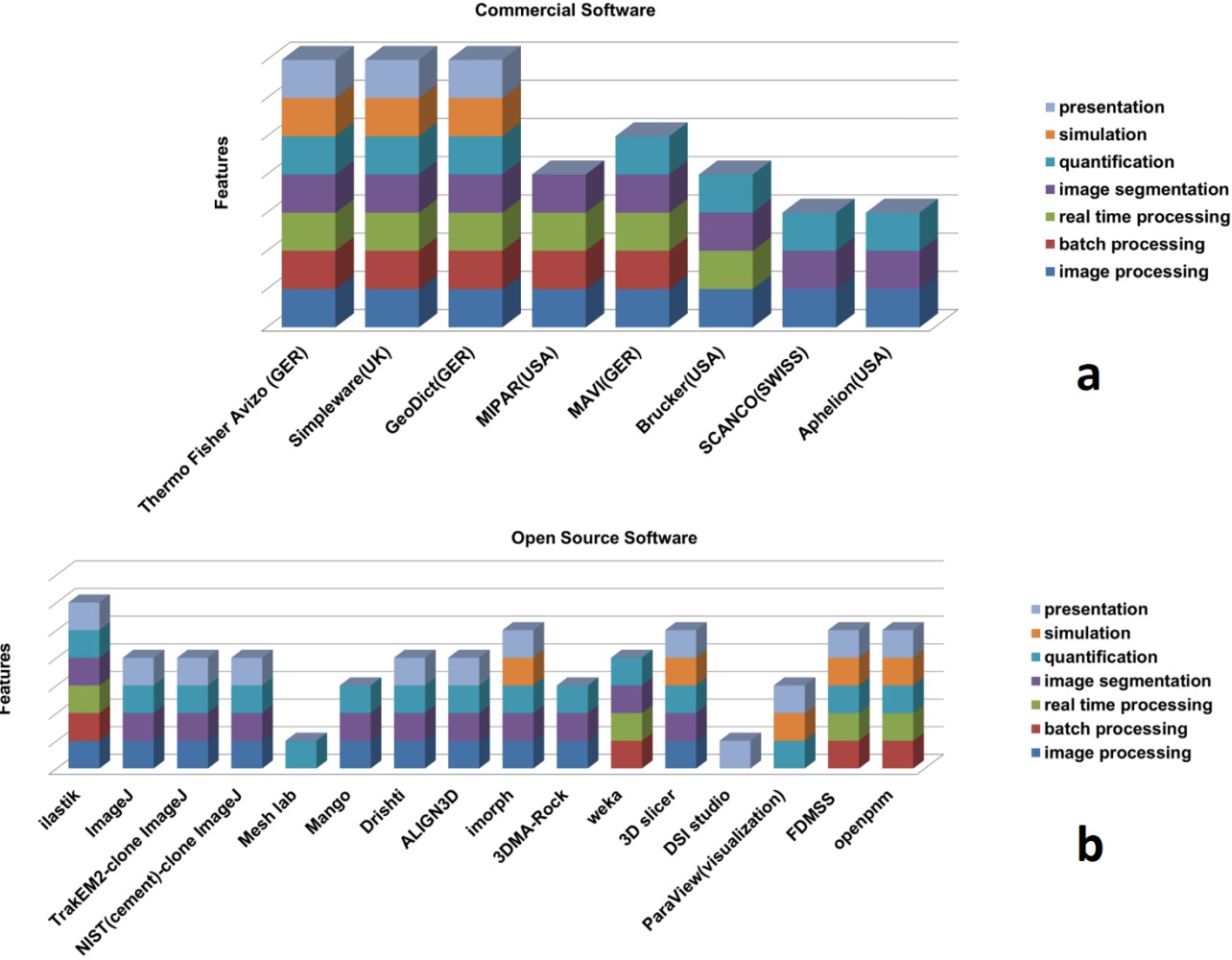

Figure 1: Market survey of the currently available commercial software (a) and open source software (b) assisting in digital rock physics analysis with features as indicated in legend

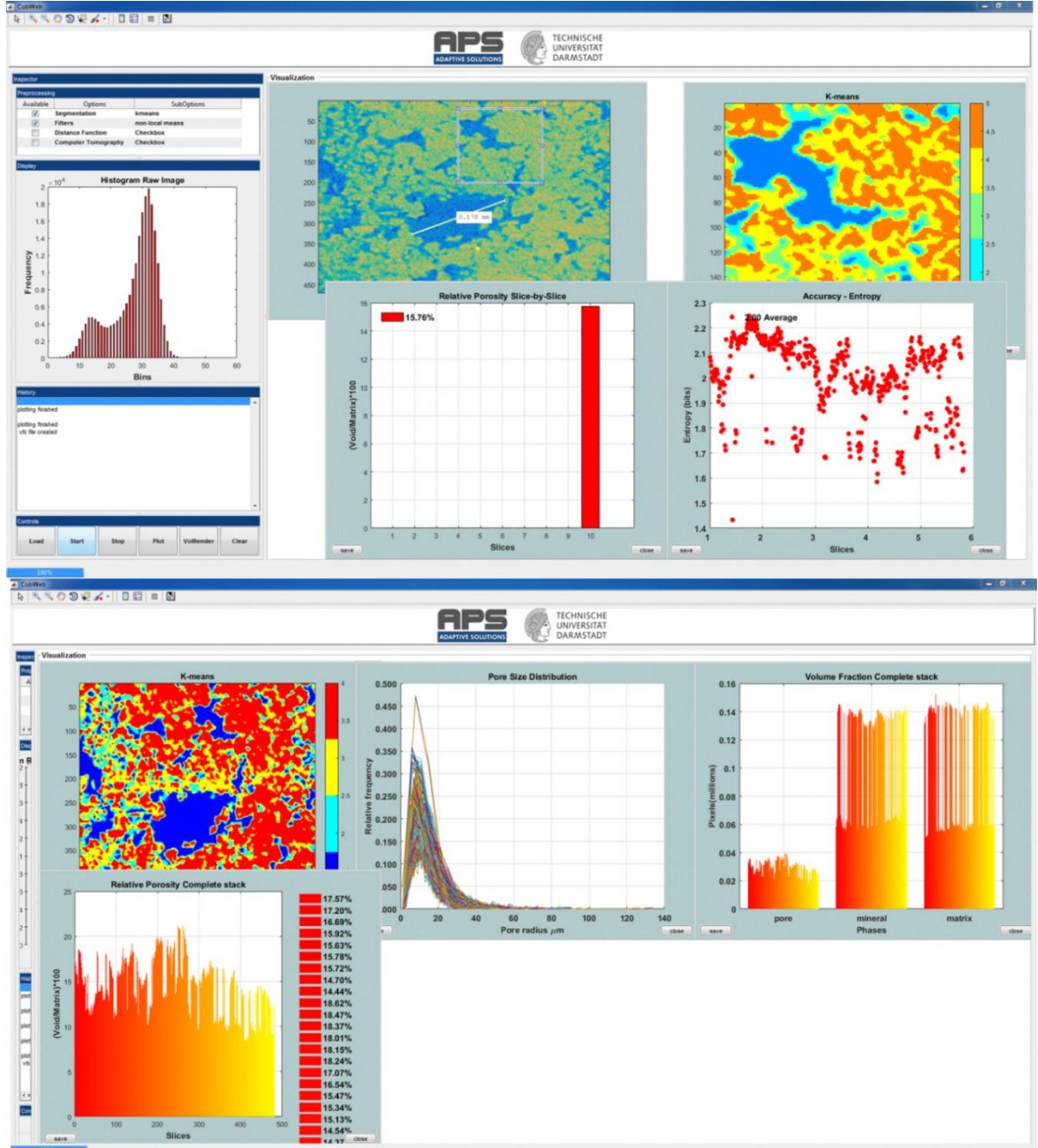

**Figure 2: Snapshots of the CobWeb GUI. XCT stack of Grosmont Carbonate rock is shown as an example for representative elementary volume analysis. The top panel displays the XCT raw sample, the K-means segmented ROI, and the porosity of single slice No. 10. The bottom plot shows pore size distribution of the complete REV stack, the relative porosity and volume fraction, respectively**

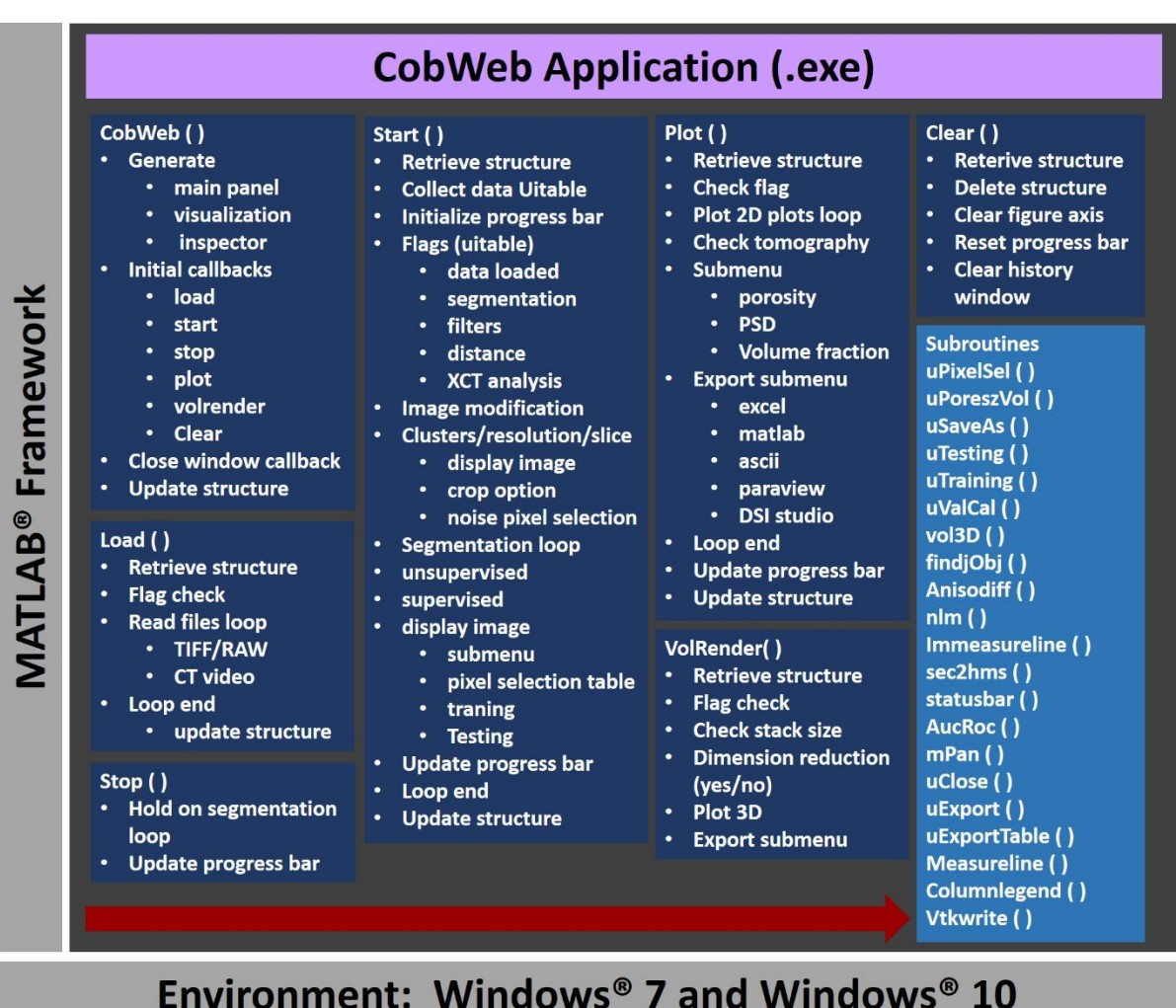

**Figure 3: The general workflow of the CobWeb software tool, where the arrow denotes the series in which different modules (represented in dark blue boxes) are compiled and executed. A separate file script is used to generate .dll binaries and executables**

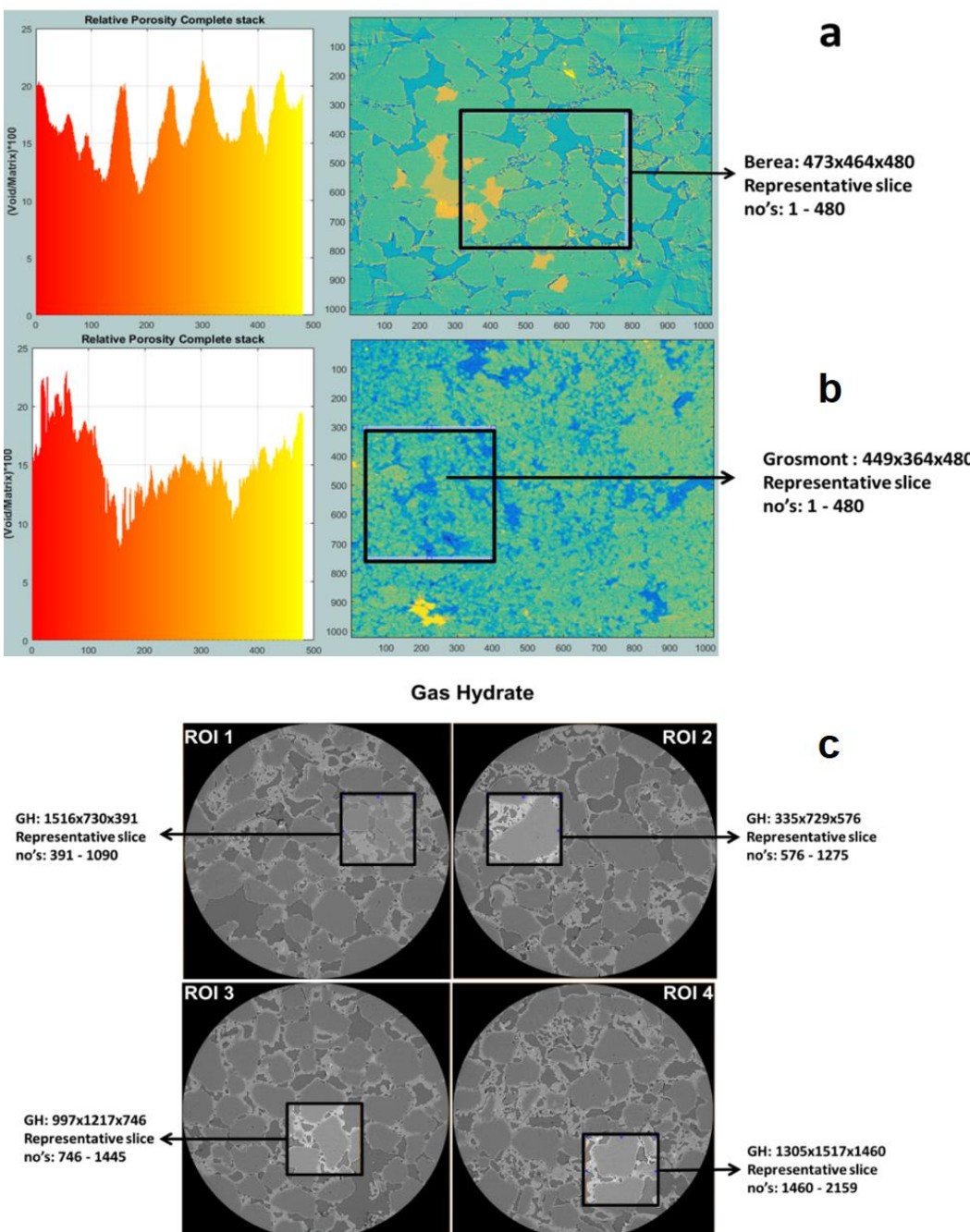

**Figure 4: The most suitable ROIs and corresponding REV dimensions of Berea sandstone and Grosmont carbonate Gas Hydrate-bearing sediment is shown in the panel a b and c respectively**

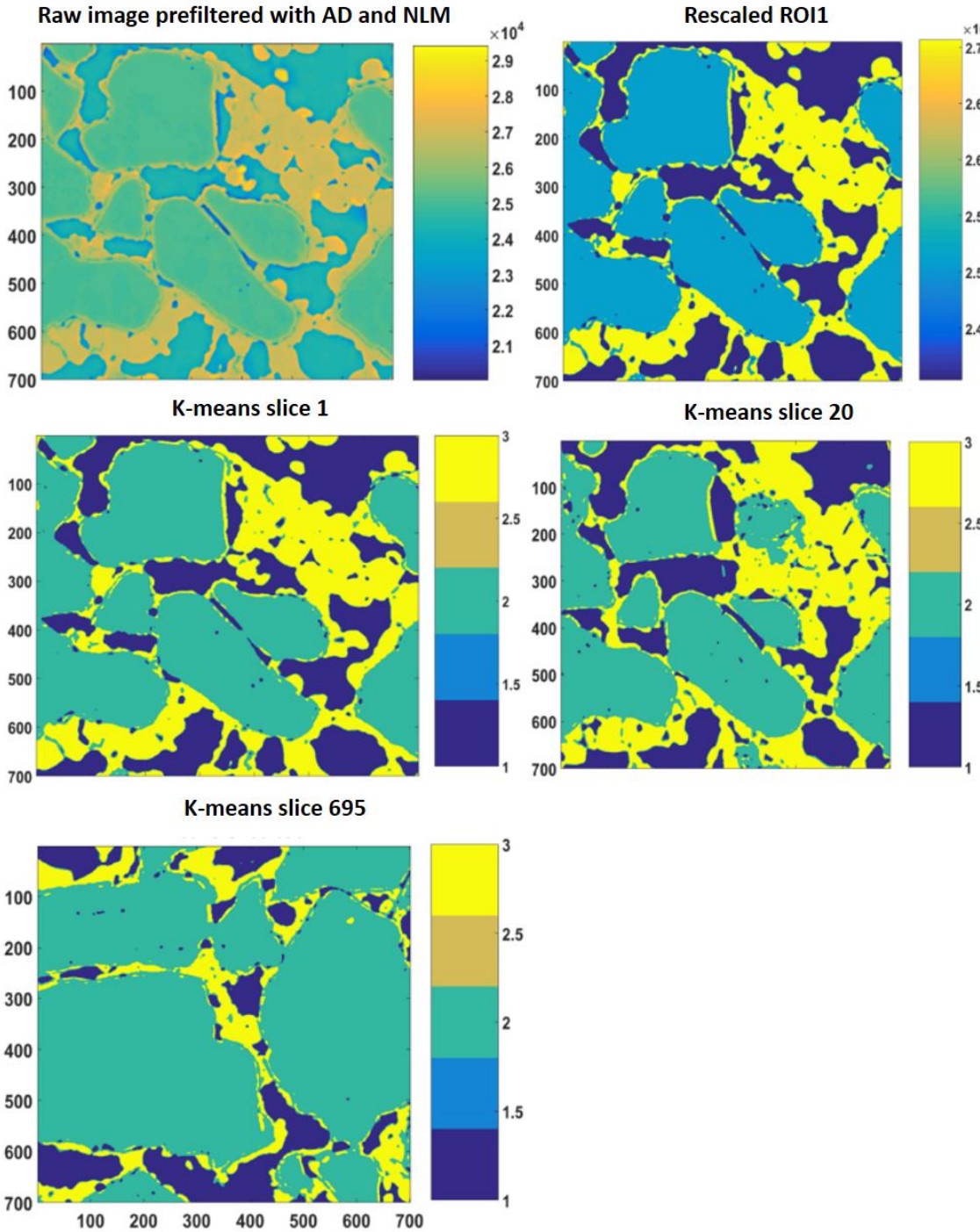

**Figure 5: 2D slices of REV 1 are represented above. The raw image is first filtered with anisotropic diffusion filtered and later on with non-local means. Thereafter, the different phases where segregated using a segmentation and indexing approach and the raw image(s) is rescaled such that they aren't any overlap or mixed phases within the raw image; and example is shown as the rescaled**

**2D ROI plot. Thereafter K-means segmentation is performed on the complete stack; 2D images of slice 1, slice 20 and slice 695 are shown as examples.**

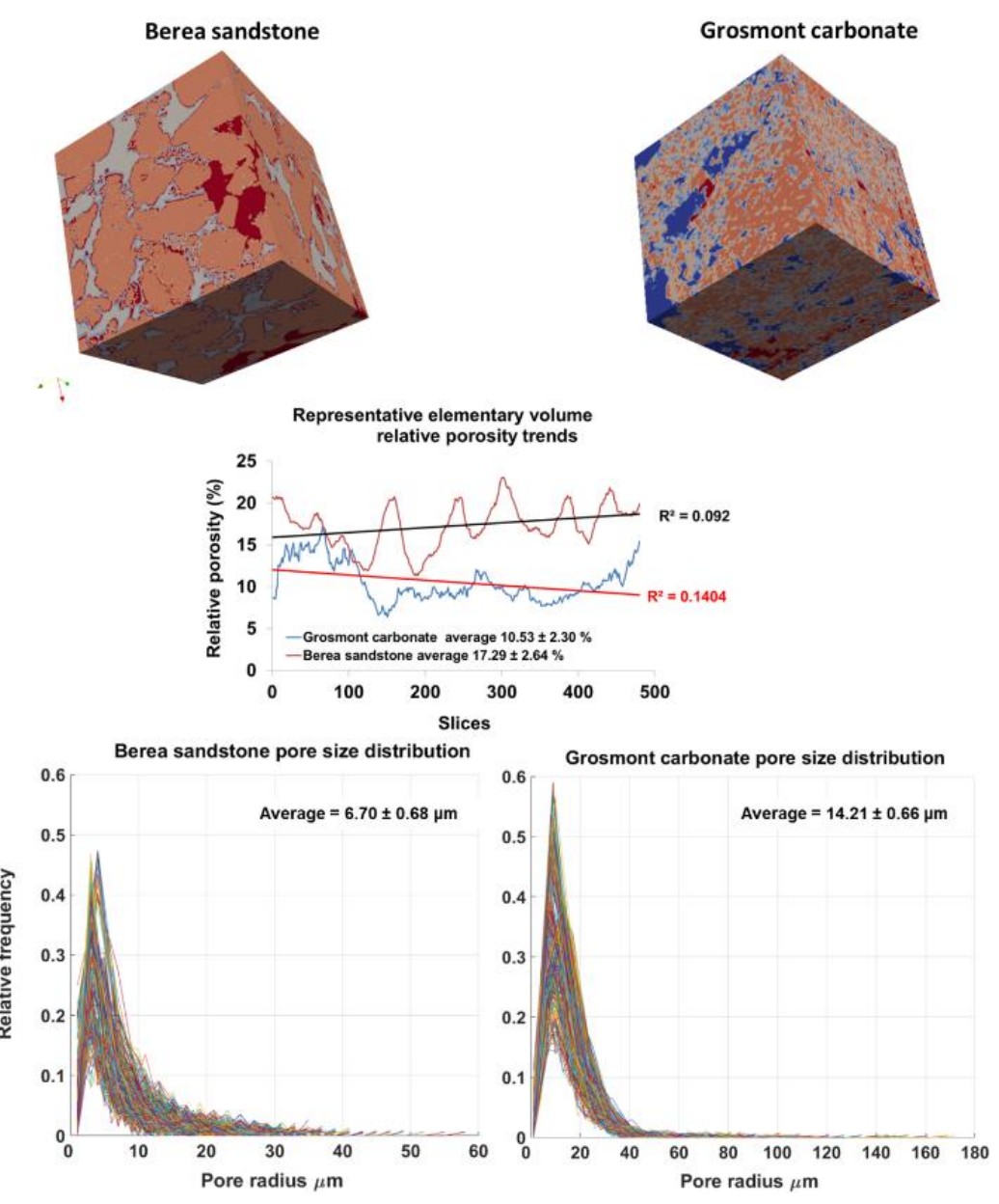

**Figure 6: Top panel shows surface plot of REVs Berea sandstone and Grosmont carbonate (size 471x478x480) using visualisation software ParaView. Middle plot shows the relative porosity (%) trend for Berea sandstone and Grosmont carbonate REVs samples. Bottom plot shows the pore size distribution of Berea sandstone and Grosmont carbonate. XCT images were segmented using K-means. In the case of Grosmont, a non-local means filter was used**

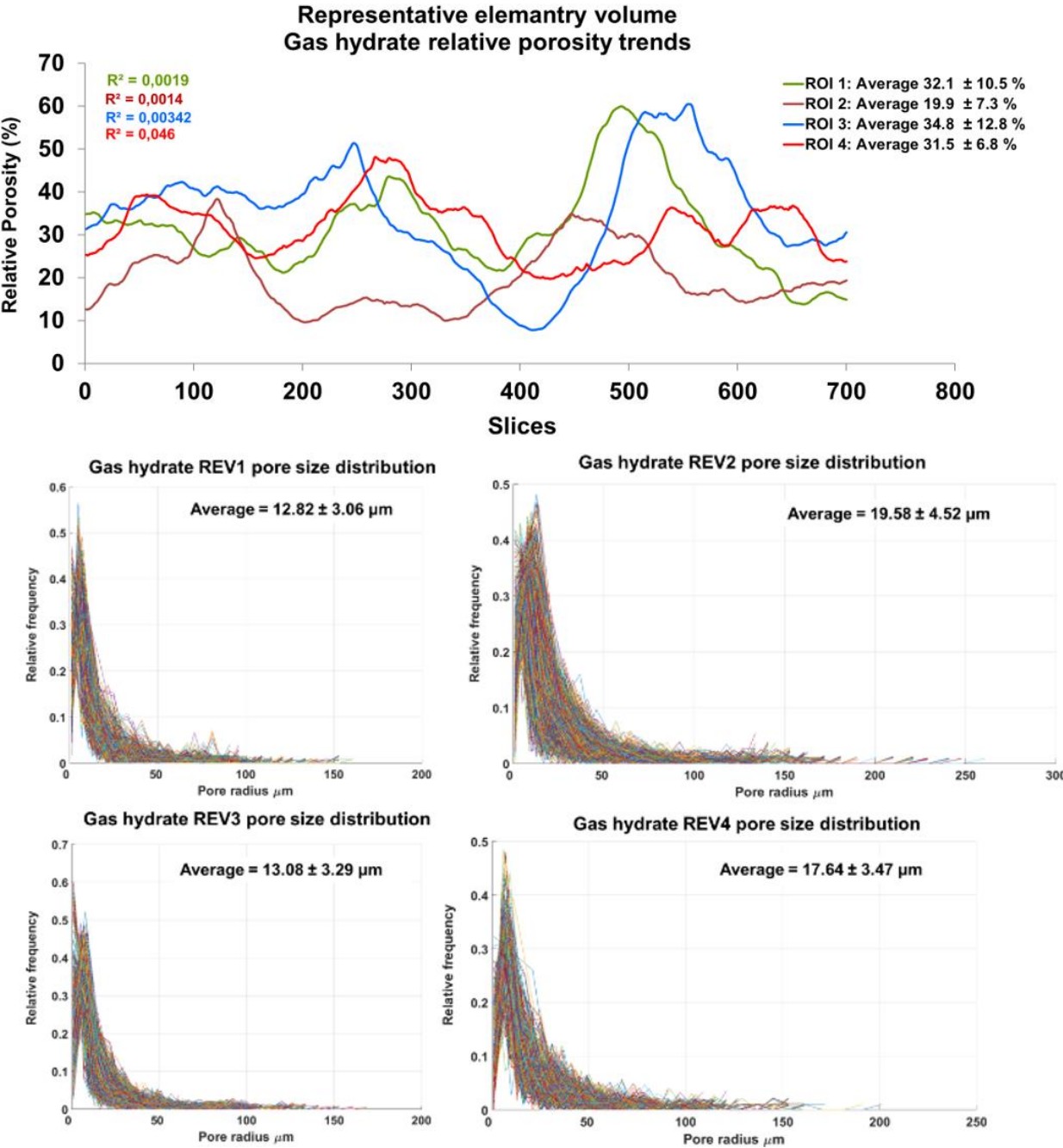

**Figure 7: The top panel shows relative porosity trend analysis of gas hydrates, the middle and bottom panel show the geometrical pore size distribution of the respective REVs. The analysis was performed using CobWeb 1.0**

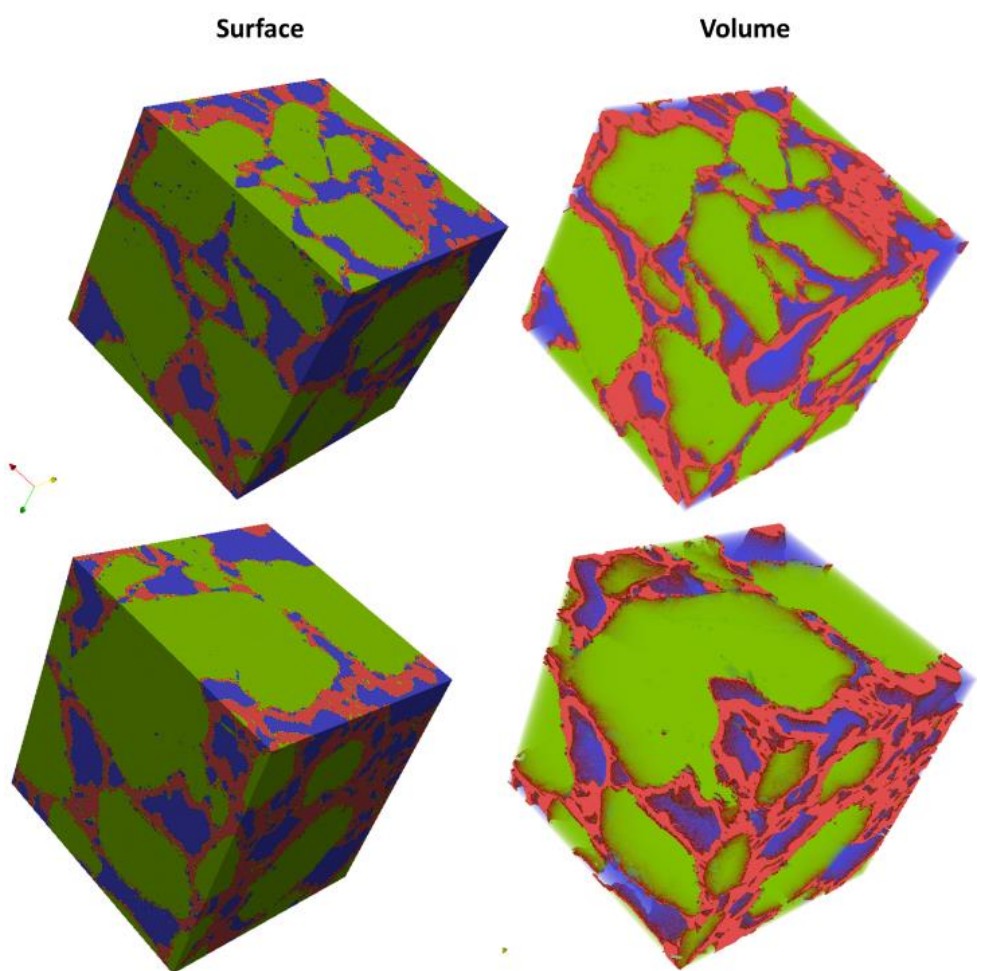

**Figure 8: Segmented REVs of a gas hydrate sample displayed as surface and volume rendered. Analyzed using CobWeb 1.0 and exported to VTK format using CobWeb 1.0 ParaView plug-in. Quartz grain phase is represented in green color, gas hydrate in red, and in blue is the void space.**

