# Peer review of "CobWeb 1.0: Machine Learning Tool Box for Tomographic Imaging"

_Geoscientific Model Development, 2018_

## Referee Comment (RC1) · Anonymous Referee #1 · 17 Apr 2019

Manuscript GMD-2018-335 presents a new collection of image processing and analyses tools for tomographic 3-D images. As the title suggests, the implementation of machine learning tools for image segmentation is in the focus of the manuscript. It is also claimed that the presented imaging software would be particularly suited for identifying representativ elementary volumes.

The connection of this paper to models is only weak, in form of the REV detection. However, the manuscript does not elaborate on how REVs are identified. It simply states that "voxel sizes around $480^3$ suited best for... ", without explaining how this conclusion was reached.

The presented software appears to be a promising piece of work, but as the authors write themselves, it still has limited capabilities. The maybe most innovative part is

implementation of machine learning routines for segmentation, albeit this in itself is not a scientific novelty. The manuscript is moreover vague when it comes to describing how the different machine learning options are implemented (with exception of the K-means clustering). It is neither explained how the cross-validation option function.

The manuscript is written in adequate English. Its structure could be improved (e.g. the description of the workings of the filters do not belong in the materials and methods). At several points I found the manuscript rather inconcise. It is e.g. not explained whether the filters and segmentation approaches work in 3-D or only in 2-D. Or like at p9L19: what choice of the cluster centers influence the perfomrance of the K-means algorithm? Its initial location? The number of clusters?

In summary, this manuscript presents a promising software tool for tomographic image analyses. But I do not think the manuscript fits within the scope of GMD, nor do I think that the manuscript is developed enough to reward revisions with another round of reviews.

I therefore recommend a release of this manuscript but encourage the authors to submit a better developed version af their paper to a better suited journal.

---

## Short Comment (SC1) · 29 Apr 2019

The archived code for this manuscript on Zenodo comprises only a binary. This does not comply with GMD's model code availability requirements, for which "code refers to computer instructions and algorithms made available as plain text". Providing only a binary provides users with no way to find out what the model code actually does. In this case, the manual indicates that the binary is tied to a particular release of Matlab, so it seems likely that the archived code will rapidly become difficult to use.

In order to be compliant, the source code needs to be properly archived. If there is a good reason why the source code cannot be archived (for example, because a third party owns the copyright and will not provide a licence), then this needs to be explicitly

stated in the code availability section.

The current binary also lacks an explicit licence, which makes it very difficult for the reader to decide what they are and are not allowed to do with the code. The Zenodo repository does have the default Zenodo licence enabled, but that is a very unusual licence choice for code, so it appears that the licence is actually just missing. This also needs to be remedied.

---

## Referee Comment (RC2) · Kirill Gerke (Referee) · 7 May 2019

The paper of Chauhan et al. presents a tool developed by the leading Author through a series of papers. This particular manuscript, unlike its predecessors, focuses on presenting the toolbox for already existing methods. I am sorry to post this review late, i guess this qualifies me as being a bad referee, but it took me quite a while to absorb everything, plus i struggled with trying to run the toolbox. On the plus side i got two reviews already posted and, thus, can rely on them to put in my vision here. I believe that topic of the paper - development of a free software to process XCT tomography images of porous media, is a very relevant topic. We do not have a solution, and those available as commercial solutions are expensive and not doing their job even closely. Scientific quality of paper is good, it is in general robust except for some particular parts

picked up by Reviewer1 and now here by Reviewer2. Scientific reproducibility was a problem for me. First, i had to borrow my wife's laptop, as mine is running under Linux. She has Matlab 2016b installed - the package is not working with it. Downloading Matlab runtime took time, as it is 1.8 Gbs of an archive. After reading the license i got puzzled - totally agree with David Ham at this point. Moreover, it was not clear how much disk space on the small SSD it will occupy - guessed it will be much more than 1.8 Gbs and i gave up. Presentation quality and writing are fine with me, i did not have any problem with these (except structure, see bellow) while reading the manuscript.

Considering the potential usage and the amount of work done by the leading Author - i believe the toolbox should be published. It can be used, cited and may serve as a comparison metric/survey of techniques. In this sense, i do not agree with the Reviewer1 who mentioned that the paper lacks novel aspects - the toolbox is the novel thing presented here (+ edge enhancement correction strategy). But i do agree with other points of critique raised by Reviewer1 and, thus, do not see how this paper could be accepted without changes. To give the Authors a possibility to publish (i think the paper is appropriate for GMD), i recommend a major revision.

Major comments: 1) REV In Section 3.3 you discuss classical REV idea of property convergence with increasing subsample volume. Yet, you do not explain how do you determine REVs for all your samples. Are these REVs were chosen in terms of porosity, pore sizes, etc.? If so, what was the threshold you used to stop increasing the volume? Moreover, here you write: "In particular, while performing permeability tensor simulation using XCT data, the size of minimum REV should be assessed not only based on porosity but also on geometrical parameters such as pore size distribution, void ratio, and coordinate number (Al-Raoush and Papadopoulos, 2010; Costanza‐Robinson M.S. et al., 2011)." I did not re-read these papers to check if these authors stated exactly this, but for performing permeability tensor simulation using XCT data, the size of minimum REV should be assessed based on permeability tensor values! More generally, REVs do not necessarily exist, finding REV for one property has nothing to do

with finding it for the other. The topic is very deep and is well beyond the scope of this manuscript. The possible ways to fix the REV problem here is to either remove it altogether, or explain for which property is it analized and calculated. By the way, if the REV analysis of porosity is based on Fig.6 and 7, i.e. by calculating porosity within separate 2D slices, it is not appropriate, it should be done within increasing volumes.

2) Structure of the methods/results Again, i totally agree with Reviewer1 and believe that current structure is hard to follow. I would suggest putting the methods description first, next describe the toolbox and functionality, then the objects and specifics of their processing. So, the structure of the paper could be something like this: - Intro as it is - Methods: 1. Image processing algorithms (here you describe all filters and segmentation techniques in better detail with references) 2. Toolbox and its functionality (you current Section 2.4) 3. Objects and specs (here you describe a sample+its processing steps, next sample, ...) - Results: 1. you current section 3.1, as well as all descriptions of filters, equations and such, belongs more to methods, not results! 2. just describe what you got for each sample 3. you do not have discussion

3) Code/toolbox The way you deliver your toolbox is not suitable for many researchers. Indeed, the code would be more useful for me. But i understand that Authors wanted a way to performs computations +deliver a GUI. And doing this with Matlab is much easier than using other tools. Simply giving away the code is not that useful for many users, because the code itself is much harder to use - you need to write a script to run something. GUI provide a possibility to use a couple of buttons and slider to do the job. Making a good GUI is very challenging. This is the reason we never released our in-house C++ image processing code with Qt GUI into public - we can use it, but for anybody else it will be buggy and not user-friendly. So, both releasing code and executables have their downsides, but in this case the need for Matlab runtime is a big obstacle for potential users. It should be relatively easy to port Matlab code into Python or Julia, which would avoid Matlab - slow and expensive dev-environment. In short, i understand why Authors decided to redistribute executables and not the code,

but still see more value in free environments (python and julia) - this does not imply that Authors have to port the code as a part of the revision.

4) You write: "The NLM filter was implemented in 3D mode to attain desired spatial and temporal accuracy and was processed on an CPU device." But Manual states that "Non-local means is only implemented as a 2D filter because filtration is done slice-by-slice".

5) Multiphase segmentations Any segmentation method can be made multiphase, be it indicator kriging, converging active contours or region growing. But the machine learning algorithms used in this paper do not solve the major problem for all multiphase segmentations - the phase having intermediate grey scale values gets sandwiched between the other two phases, e.g., see numerous fugues in the paper and in the manual. I would like you to elaborate on this and why do you think clustering and k-means approaches are better than aforementioned techniques?

6) Verification In the Section 6.2 in the Manual your provide validation metrics to prove your segmentation results. I do not see how any of these are actually do the job! They are simply the metrics for performance of the machine learning algorithms, not the segmentation results. I strongly suggest remove or re-write this part. We technically do not have verification data - the ideal segmentations, which can be produced only using synthetic tomography. You could discuss these things a little bit, so that comments 5-6) will give material for small discussion subsection you lack at the moment.

Minor comments: 1) In introduction you write: "Our hypothesis is that 3D tomographic REV analysis of Berea SandstoneTM (BS), Grosmont carbonate rock (GCR), and gas hydrate (GH)-bearing sediment datasets, would benefit of this new approach." I find this statement scientifically awkward (even without considering the REV problems in this paper), moreover this hypothesis is nowhere to be tested. 2) Section 3.1.1.2, you write: "In this study, the NLM filter was set to a search window of 21, local neighbour-hood of 6 and a similarity value of 0.71." As i was unable to test the toolbox, it seems

after reading the Manual that too many things are fixed. How useful is it going to be if these NLM and other parameters are not changable within the toolbox? With the parameters you specify from my experience with NLM (our version is different from yours and follows closely Buades and Morel paper) i would expect the results to be kind of oversmoothed (again, i did not run the toolbox, so this is a guess). 3) As far as i understood, technically you do not compute pore-size distribution (e.g., extract the pore-network model), but distance-map transform and analyse its distribution. If so, please, change the description of the methodology accordingly. Moreover, i do not understand the legends on the graphs, be it fig.8 in the Manual or fig.8 in the paper (what are these numerous pore-size distributions for the same sample?). 4) You mention a number of free software solutions. I strongly suggest adding our recent effort - FDMSS software to simulate single-phase flow within 3D pore geometries into your list of tools that could be used after segmentation of 3D images using CobWeb. Moreover, you could compute permeabilities for your segmentations and compare then against available data, e.g., "Laboratory measurements of porosity and permeability reported by (Andrä et al., 2013b) are around 21 % ($\varphi = 0.21$) and $\kappa$ = 150 mD âĬĂ 470 mD, respectively".

This review got too long. I appreciate the efforts of the Authors to produce this toolbox and strongly support its publication with GDM. For these reasons i was trying to provide all possible solutions for current manuscript's problems.

---

## Author Comment (AC2) · 31 May 2019

Dear Dr. David Ham,

Following are answers to your comments.

Sincerely,

Swarup Chauhan, Kathleen Sell, Frieder Enzmann, Wolfram Rühaak, Thorsten Wille, Ingo Sass, Michael Kersten.

The reviewer comments are formatted in italics and the authors response to the comments are formatted in bold.

 Notation *SC1.P#* represents ReviewersComment.ParagraphNumber

*SC1.P1 The archived code for this manuscript on Zenodo comprises only a binary. This does not comply with GMD's model code availability requirements, for which "code refers to computer instructions and algorithms made available as plain text". Providing only a binary provides users with no way to find out what the model code actually does. In this case, the manual indicates that the binary is tied to a particular release of Matlab, so it seems likely that the archived code will rapidly become difficult to use.*

**As mentioned in the conclusions, page 12 line 26 onwards, CobWeb is been and will be developed further. The current version of CobWeb requires compiler version 2017b. We don't think, that the archived code will be outdated; MathWorks® archives compilers as old as R2012a (7.17).**

*SC2.P2 In order to be compliant, the source code needs to be properly archived. If there is a good reason why the source code cannot be archived (for example, because a third party owns the copyright and will not provide a licence), then this needs to be explicitly stated in the code availability section.*

**CobWeb relies on certain external libraries such as LS-SVMlab software for LSSVM segmentation (**[https://www.esat.kuleuven.be/sista/lssvmlab/](https://www.esat.kuleuven.be/sista/lssvmlab/)**) and MathWorks® internal machine learning libraries which are available for non-commercial purpose and may not be used for commercial purpose without an explicit written permission. Therefore, as mentioned in the reply to reviewer 1 (***RC1.P3.***), we are in a decision phase and are considering the licence issues. But, we certainly can provide the source code to the reviewers for evaluation.**

**Through the manuscript we are hoping that the scientific community will take notice and will lead to further inputs, collaborations or possible benchmark studies.**

*SC3.P2 The current binary also lacks an explicit licence, which makes it very difficult for the reader to decide what they are and are not allowed to do with the code. The Zenodo repository does have the default Zenodo licence enabled, but that is a very unusual licence choice for code, so it appears that the licence is actually just missing. This also needs to be remedied.*

**Thanks for pointing it out. We will include a license file, wherein the GUI can be used for scientific studies under**

**"Creative Commons License Version 4.0", available at**
http://creativecommons.org/licenses/by-nc-sa/4.0/

---

## Author Response (AR1)

Dear Dr. Thomas Poulet,

Today, I have uploaded a revised version of our manuscript gmd-2018-335 title "CobWeb 1.0: Machine Learning Tool Box for Tomographic Imaging". As mentioned in our replies to the respective referees, we have implemented all the changes. We think these updates to the manuscripts have considerably improved it quality.

In a rough overview the updates are as follows (more details can be found in the referee-replies and respective changes in the manuscript):

1) Both referees, emphasized the structure of the manuscript was not suitable and referee 2 suggested a scheme to structure the manuscript and further pointed an absence of discussion section. This suggestion was taken into consideration and the structure has been changed accordingly.

2) Referee 2 found, that the REV section was out of the scope of the study. We agree to his comment and have removed the section. Both the referees enquired about the selection procedure of the REV and segmentation scheme. In the result and discussion section, subsection 5.1. (data selection) elaborated on the REV selection and subsection 5.2 (data processing) discusses the 2D slice by slice segmentation.

3) Referee 2 suggested that the role of ML to improve multiphase segmentation needs to be addressed. This is included in section result and discussion subsection 5.3 (multiphase image segmentation), where suggestions are given to preserve the intermediate phase, which gets misclassified. As enquired by referee 1, the role of placing centroid centers in improving the performance (accuracy and speed) of ML techniques has been addressed in subsect 5.3. (multiphase image segmentation).

4) As pointed by referee 2, Figure 1 has been updated; CobWeb user manual page 55 section 6.2 (Validation), following line "The validation analysis is not a direct validation of segmentation but is rather to verify the performance of the machine learning algorithms" has been included for clarification. License file has been included. The Introduction section, includes relevance of CobWeb for the XCT community (reply to referee 1 comments RC1.P3).

In the manuscript, the updated passages are colored in blue and the deleted lines are strikethrough in red.

Best Regards

Swarup Chauhan

We thank the referee for the time and effort to review our manuscript. Below, we give point-by-point response to your concern.

Sincerely,

Swarup Chauhan, Kathleen Sell, Frieder Enzmann, Wolfram Rühaak, Thorsten Wille, Ingo Sass, Michael Kersten.

Anonymous Referee #1

The reviewer comments are formatted in italics and the authors response to the comments are formatted in bold.

Notation *RC1.P#* represents ReviewersComment.ParagraphNumber

*RC1.P1 Manuscript GMD-2018-335 presents a new collection of image processing and analyses tools for tomographic 3-D images. As the title suggests, the implementation of machine learning tools for image segmentation is in the focus of the manuscript. It is also claimed that the presented imaging software would be particularly suited for identifying representativ elementary volumes.*

**Yes. Apart from that, the manuscript also highlights a new procedure termed as, dual filtering and dual segmentation to remove edge enhancement artefact in synchrotron based images using machine learning approach. To our knowledge this hasn't been published before.**

**The scientific community will benefit from the novelty of this approach. The code has been made available. We can certainly elaborate this in the introduction of the manuscript.**

**Not revised**

*RC1.P2 The connection of this paper to models is only weak, in form of the REV detection. However, the manuscript does not elaborate on how REVs are identified. It simply states that "voxel sizes around 480ˆ3 suited best for... ", without explaining how this conclusion was reached.*

**OK. In the revised manuscript we will extend the section 3.3 and elaborate on the identification of REV for all the three samples.**

**Basically, it was a combination of visual inspection and consecutively segmenting and plotting tends in relative porosity, pore size distribution and volume fraction. This was done by loading the complete stack in the CobWeb software, during the loading process a 2D movie of the tomogram is displayed in the display window and saved in the root folder. Carefully monitoring the movie gives an objective evaluation of the heterogeneity of the respective XCT sample. Thereafter, based on this subjective information different ROIs are selected, cropped, segmented and their respective geometrical parameter are intercompared.**

**In the case of Berea sandstone, four different ROIs were investigated, whereas Grosmout carbonate rock seven different ROIs where need to identify the best REVs. Through our previous scientific studies on the GH sediments (Sell et al., 2016; Sell et al., 2018) we were aware or best-suited REVs. The identification of best REV for Grosmout was relatively tedious compared to Berea sandstone and GH sediment; due to the low resolution and microporosity present in the Grosmount tomograms.**

**Revised page 18 | line 13 - 23**

The intention of showing particularly only two REV trends of relative porosity in Figure 7. is due to a very good agreement in porosity values to the benchmark publication of Andrä et al., (2013a, 2013b).

*RC1.P3. The presented software appears to be a promising piece of work, but as the authors write themselves, it still has limited capabilities. The maybe most innovative part is implementation of machine learning routines for segmentation, albeit this in itself is not a scientific novelty.*

The software is built on scientific studies which have been peer-reviewed and accepted in the scientific community Chauhan et al., 2016a,b. The spinoff for these studies was not the lack of accuracy provided by manual segmentation schemes, but the subjective assessment and non-comparability caused by the individual human assessments. Therefore, the automated segmentation schemes offer speed, accuracy and possibility to intercompare results, enhancing traceability and reproducibility in the evaluation process. To our knowledge none of the XCT software used in rock science community relies on machine learning for segmentation explicitly, which makes the software unique if not novel.

Despite many review articles and scientific publication highlight potential of machine learning and deep learning (Iassonov et al., 2009; Cnudde and Boone, 2013; Schlüter Steffen et al., 2014), software libraries or toolbox are seldom made available. Thus, with CobWeb we started for the first time to fill this gap, and despite its limited volume rendering capabilities— it is a useful tool and current version of the software can be applied in scientific and industrial studies. Certainly a conscious decision need to be taken on our side if to dedicate CobWeb as a segmentation tool or expand it towards simulation software like MATH2MARKET, GeoDict or Volume Graphics. On the other side CobWeb provides an appropriate test platform, where new segmentation and filtration schemes can be tested and used as a complementary tool to the simulation software GeoDict and Volume Graphics. The simulation softwares (GeoDict and Volume Graphics) have benchmarked solvers for performing flow, diffusion, dispersion, advection type simulation, but their accuracy relies heavily on the finely segmented datasets.

**Revised page 2 | line 11-26**

*RC1.P3 The manuscript is moreover vague when it comes to describing how the different machine learning options are implemented (with exception of the K-means clustering). It is neither explained how the cross-validation option function.*

We acknowledge reviewers concern. But, in section 2.4.2 (page 5) we have cited Chauhan et al., (2016a); Chauhan et al., (2016b) which covers the details about the algorithms and cross-validation schemes. Also, Figure 3 gives a visual overview how the ML techniques fit into the framework. Thereafter, the user manual published on the zendo repository (https://dx.doi.org/10.5281/zendo.2390943)
explains the implementation and how to use the segmentation algorithms. Within the scope of the manuscript we find the description sufficient.

But if required, we will expand further on the implementation of the ML techniques and the cross-validation options.

*RC1.P4 The manuscript is written in adequate English. Its structure could be improved (e.g. the description of the workings of the filters do not belong in the materials and methods).*

**Ok. The description and working of filters is written under the sub section 3.1 image processing not under materials and methods.**

*RC1.P4 At several points I found the manuscript rather inconcise. It is e.g. not explained whether the filters and segmentation approaches work in 3-D or only in 2-D.*

**We thank the reviewer for pointing it out. The CobWeb 1.0 uses a slice-by-slice 2D approach. It was observed that the ML techniques tend to underestimate porosity values compared to manually segmented analysis at an REV scale size > 500³. This substantial degree of uncertainty is caused due to 2D slice-by-slice processing rather than the ML techniques. The 2D slice-by-slice approach, passes only, the spatial information (X, Y coordinate direction) to the ML algorithms, which ends up sorting the intensity variation in the spatial domain (local maxima). Therefore, the lack of temporal information (Z coordinate direction) restricts the degree of freedom to find at a global spatial-temporal optimum. In other words, as the temporal changes arise, due to bedding (sedimentary rock) or micro porosity (carbonate rocks) in the rock texture, they are represented as sudden spike or dip in porosity values; which to an inexperienced eye appear as artefact or anomalies– and often-then-not discarded.**

**This correction will be implemented in the next software version; in the current workflow it has not been accounted for (CobWeb 1.0). Since, it requires refactoring the loop-based scalar-oriented framework to matrix and vector operation approach called *vectorization*. The 2D slice-by-slice processing scheme is much faster compared to the 3D approach. So, the choice of 2D processing for this research study was made to make it affordable to compute on desktop, laptop for near real-time and onsite evaluation.**

*RC1.P4 Or like at p9L19: what choice of the cluster centers influence the perfomrance of the K-means algorithm? Its initial location? The number of clusters?*

**Thanks again, for raising the question, we can certainly elaborate on this in the discussion section. In general, performance in terms of accuracy and speed is directly proportional to starting point (initial location) in the segmentation process. Meaning, the closer the starting point (initial location) is to the**

global minima— faster will the algorithm converge and even so better is the performance (accuracy & speed).

However, in unsupervised technique by default the choice of the starting point is through random seed unless explicitly specified. So, in the case of the dual segmentation approach, the intuition was to capture all the material phases, including the edge enhancement artefact, speck and noise etc. in the first step and thereafter in the second step to rescale them to the plausible phases.

 Hence, in the first step the 20 clusters where initialized using random seed. Since, the priority was to capture all phases in GH tomograms not the performance. And, after the rescaling processes, we were aware of the initial locations which we used as starting point (initial location) to assist the algorithm to move towards identifying correct phases.

**Revised: Page 17 | line 21 - 33**

*RC1.P5 In summary, this manuscript presents a promising software tool for tomographic image analyses.* **We thank the reviewer for the acknowledgement.**

*RC1.P5 But I do not think the manuscript fits within the scope of GMD, nor do I think that the manuscript is developed enough to reward revisions with another round of reviews.*

**We disagree with the reviewer on the above comment. The uniqueness of this journal is that, it gives the possibility of accepting six different types of manuscripts, and we were careful in placing the work in the *model description papers* category as it fulfils most of its norms if not all. This has been clearly highlighted in the cover letter to the topical editor.**

Dear Dr Kirill Gerke,

Thanks for your time, praise for our work and useful comments. Your comments and suggestions gave us some fresh ideas and a new perspective on the topic and helped us to improve our manuscript. Please find below a point-by-point response to your concerns.

The reviewer comments are formatted in italics and the author's response to the comments are formatted in bold.

Notation *RC2.P#* represents Reviewers Comment. Paragraph Number

*RC1.P1: The paper of Chauhan et al. presents a tool developed by the leading author through a series of papers. This particular manuscript, unlike its predecessors, focuses on presenting the toolbox for already existing methods. I am sorry to post this review late, i guess this qualifies me as being a bad referee, but it took me quite a while to absorb everything, plus I struggled with trying to run the toolbox. On the plus side I got two reviews already posted and, thus, can rely on them to put in my vision here. I believe that topic of the paper - development of a free software to process XCT tomography images of porous media, is a very relevant topic. We do not have a solution, and those available as commercial solutions are expensive and not doing their job even closely. Scientific quality of paper is good, it is in general robust except for some particular parts*

**Thanks for the trust and the praise.**

*RC1.P1: Scientific quality of paper is good, it is in general robust except for some particular parts picked up by Reviewer1 and now here by Reviewer2. Scientific reproducibility was a problem for me. First, i had to borrow my wife's laptop, as mine is running under Linux. She has Matlab 2016b installed - the package is not working with it. Downloading Matlab runtime took time, as it is 1.8 Gbs of an archive. After reading the license i got puzzled - totally agree with David Ham at this point. Moreover, it was not clear how much disk space on the small SSD it will occupy - guessed it will be much more than 1.8 Gbs and i gave up. Presentation quality and writing are fine with me, i did not have any problem with these (except structure, see below) while reading the manuscript.*

**A possible solution is that we can send you the source code with the make file, so you may generate the GUI and/or test the source code in the Matlab environment. As mentioned in the code availability section (page 13 line 10) the GUI requires Matlab runtime compiler R2017b (9.3).**

**As addressed in the short comments from Dr David Ham we will include an appropriate licence file in the zendo repository.**

**Revised: Updated license file**

*Major comments:*

*1) REV In Section 3.3 you discuss classical REV idea of property convergence with increasing subsample volume. Yet, you do not explain how do you determine REVs for all your samples. Are these REVs were chosen in terms of porosity, pore sizes, etc.? If so, what was the threshold you used to stop increasing the volume?*

**As explained in the reply to the comments from reviewer 1 (RC1.P2): Basically, it was a combination of visual inspection and consecutively segmenting and plotting trends in relative porosity, pore size**

distribution and volume fraction. We gradually increase the sub-sample volume at a different location (X, Y) and depth (Z) inside the XCT tomograms. And, for each subsample volume, the geometrical parameters are intercompared.

The main indicator, however, was the porosity trend; i.e. when regression coefficient R2 was close to zero, it was an indicator that the sub-volume has accumulated the heterogeneity along the z-axis of the sample. Therefore, based on the trend analysis approach, the sub-volume dimension where R2 value was close to zero was chosen as the suitable REV.

**Revised: page 15 | line 13 – 30.**

*Moreover, here you write: "In particular, while performing permeability tensor simulation using XCT data, the size of minimum REV should be assessed not only based on porosity but also on geometrical parameters such as pore size distribution, void ratio, and coordinate number (Al-Raoush and Papadopoulos, 2010; Costanzâˇ AˇRRobinson M.S. et al., 2011)." I did not re-read these papers to check if these authors stated exactly this, but for performing permeability tensor simulation using XCT data, the size of minimum REV should be assessed based on permeability tensor values! More generally, REVs do not necessarily exist, finding REV for one property has nothing to do with finding it for the other. The topic is very deep and is well beyond the scope of this manuscript. The possible ways to fix the REV problem here is to either remove it altogether, or explain for which property is it analized and calculated.*

**Thanks for the suggestion, The section 3.3 and the respective figure 6 has been removed P11|line 4-21.**

**Removed P11 | line 4-21.**

*By the way, if the REV analysis of porosity is based on Fig.6 and 7, i.e. by calculating porosity within separate 2D slices, it is not appropriate, it should be done within increasing volumes.*

**In, the current version of CobWeb the porosity is calculated for 2D slices. The 2D slice-by-slice approach incorporated in CobWeb may lead to some uncertainties as the porosity variation is only accounted for along the Z-axis. However, by comparing different subsamples and accurately isolating (segmenting) the pore phase and calculating the mean porosity of the complete stack these fluctuations can be normalized.**

**As a consequence of your suggestion, now the porosity is calculated for a complete 3D stack. By indexing the pore phase pixels for the 3D stack and calculating the ratio between the pore phase pixel and matrix phase.**

**Not revised**

*2) Structure of the methods/results Again, i totally agree with Reviewer1 and believe that current structure is hard to follow. I would suggest putting the methods description first, next describe the toolbox and functionality, then the objects and specifics of their processing. So, the structure of the paper could be something like this: - Intro as it is - Methods: 1. Image processing algorithms (here you describe all filters and segmentation techniques in better detail with references) 2. Toolbox and its functionality (your current Section 2.4) 3. Objects and specs (here you describe a sample+its processing steps, next sample, ...) - Results: 1. you current section 3.1, as well as all descriptions of filters, equations and such, belongs more to methods, not results! 2. just describe what you got for each sample 3. you do not have discussion*

We thank the reviewer for this suggestions. We have implemented it accordingly in the revised manuscript.

**Revised:**

**2. Image processing subsection page 3 | line 11-17; page 5 | line 1-23**

**Section 2.2 Image Segmentation page 4 | line 24 to – to Page 9 | line 19**

**Section 2.3 Performance page 7 | line 19 – to page 9 | line 1**

*3) Code/toolbox The way you deliver your toolbox is not suitable for many researchers. Indeed, the code would be more useful for me. But i understand that Authors wanted a way to performs computations +deliver a GUI. And doing this with Matlab is much easier than using other tools. Simply giving away the code is not that useful for many users, because the code itself is much harder to use - you need to write a script to run something. GUI provide a possibility to use a couple of buttons and slider to do the job. Making a good GUI is very challenging. This is the reason we never released our in-house C++ image processing code with Qt GUI into public - we can use it, but for anybody else it will be buggy and not user-friendly. So, both releasing code and executables have their downsides, but in this case the need for Matlab runtime is a big obstacle for potential users. It should be relatively easy to port Matlab code into Python or Julia, which would avoid Matlab - slow and expensive dev-environment. In short, i understand why Authors decided to redistribute executables and not the code, but still see more value in free environments (python and julia) - this does not imply that Authors have to port the code as a part of the revision.*

**We thank the reviewer for his support and understanding. Despite the easy semantics provided by Matlab, development of the GUI in Matlab was not easy. We used tips and tricks from undocumented Matlab by (https://undocumentedmatlab.com/). The idea was to make it available for many researchers of the science communities, therefore GUI was a feasible option.**

4) You write: "The NLM filter was implemented in 3D mode to attain desired spatial and temporal accuracy and was processed on a CPU device." But Manual states that "Non-local means is only implemented as a 2D filter because filtration is done slice-by-slice".

**Yes. The NLM filter is hard-coded as 2D in the CobWeb standalone version (GUI). But, by tweaking or modifying the source code we could initially pre-processed the XCT images using NLM 3D filtration and thereafter subjected it to segmentation.**

**The information given above will be added to the discussion section and will also be highlighted in the conclusions**

**Revised page 19 | line 20-23**

5) Multiphase segmentations

Any segmentation method can be made multiphase, be it indicator kriging, converging active contours or region growing. But machine learning algorithms used in this paper do not solve the major problem for all multiphase segmentations - the phase having intermediate grey scale values gets sandwiched between the other two phases, e.g., see numerous fugues in the paper and in the manual. I would like you to

elaborate on this and why do you think clustering and k-means approaches are better than aforementioned techniques?

**Thanks for your suggestion, certainly we have elaborate on this in the manuscript in the discussion section.**

**In a practical sense, machine learning tries to separate grayscale values in to disjoint sets. The creation of these disjoint sets can be created in two ways.**

**1) By binning the greyscale values to the nearest representative values which is iteratively updated using an optimization function. This optimization function can be a simple regression or distance function (Jain et al. 1999), commonly used in the unsupervised techniques.**

**2) Or by regularizing pre-trained models which store certain pattern information of the datasets such as topology features, contour intensities, pixel value etc. (Hopfield 1982; Haykin 1995; Suykens and Vandewalle 1999). Or by using a voting system in a bootstrap ensemble of linear models (Breiman 1996).**

**So during this process, the intermediate greyscale values which have low sample size are merged in to larger sets of a high sample size to create disjoint boundaries.**

**One way to overcome this problem is by using supervised techniques such as LSSVM or Ensemble classifiers. When constructing a training dataset (feature vector selection), careful selection of intermediate phases as a sufficiently large sample size compared to the predominate phases will preserve the intermediate phases. And, the likelihood that the trained model will identify them and cluster them separately is higher (Chauhan et al. 2016).**

**In this study, in particular, we made tests using supervised techniques (LSSVM, Ensemble classifies) and unsupervised technique (FCM) but the results weren't superior compared to K-means. Therefore we choose K-means as it was faster compared to other ML techniques.**

**Revised: page 17 | line 1-22**

6) Verification In the Section 6.2 in the Manual your provide validation metrics to prove your segmentation results. I do not see how any of these are actually do the job! They are simply the metrics for performance of the machine learning algorithms, not the segmentation results. I strongly suggest remove or re-write this part.

**We agree to the reviewers views. In the revised version of the manual it is rewritten that these are simply the metrics for performance of the machine learning algorithms not validation.**

**Revised: CobWeb manual page 55**

We technically do not have verification data - the ideal segmentations, which can be produced only using synthetic tomography. You could discuss these things a little bit, so that comments 5-6) will give material for small discussion subsection you lack at the moment.

**We thank the reviewers for his suggestion. We agree and it will incooperate it in the discussion section.**

**Revised: CobWeb manual page 55**

Minor comments:

1) In introduction you write: "Our hypothesis is that 3D tomographic REV analysis of Berea SandstoneTM (BS), Grosmont carbonate rock (GCR), and gas hydrate (GH)-bearing sediment datasets, would benefit of this new approach." I find this statement scientifically awkward (even without considering the REV problems in this paper), moreover this hypothesis is nowhere to be tested.

**As per reviewers suggestion, Page 1 line 15- 17 "Our hypothesis is that 3D tomographic REV analysis of Berea SandstoneTM (BS), Grosmont carbonate rock (GCR), and gas hydrate (GH)-bearing sediment datasets, would benefit of this new approach." has been removed from the manuscript**

2) Section 3.1.1.2, you write: "In this study, the NLM filter was set to a search window of 21, local neighbourhood of 6 and a similarity value of 0.71." As i was unable to test the toolbox, it seems after reading the Manual that too many things are fixed. How useful is it going to be if these NLM and other parameters are not changeable within the toolbox?

**Data prepossessing is a very important step before segmentation. Depending on the resolution of the dataset, the fixed parameters of NLM and other filters should do a fairly good job. In case, their still exists noise and artefacts it is recommended to use supervised techniques. The residual noise or artefact pixel values can be captured through proper feature vector selection, and their after training the appropriate model and performing classification. Thereby, the existing noise and artefact can be isolated and segmented as separate labels.**

**Another alternative option could be to pre-process the data with desired filters data and imported the data into CobWeb for segmentation and analysis.**

**Revised: page 16 | line 9-17**

3) As far as i understood, technically you do not compute pore-size distribution (e.g., extract the pore-network model), but distance-map transform and analyse its distribution. If so, please, change the description of the methodology accordingly. Moreover, i do not understand the legends on the graphs, be it fig.8 in the Manual or fig.8 in the paper (what are these numerous pore-size distributions for the same sample?).

**Yes, It will be revised in the manuscript.**

**As mentioned in (Rabbani et al. 2014), the aim is to breakdown the monolithic void structure of rock into specific pores and throats connecting to each other. Rabbani et al. (2014) used image processing algorithm (median filter) to perform pre-processing. In our case, the tomograms have already been pre-processed and segmented through ML techniques and thereafter subjected it to the watershed algorithm.**

**Page 18 | line 3 - 10**

4) You mention a number of free software solutions. I strongly suggest adding our recent effort FDMSS software to simulate single-phase flow within 3D pore geometries into your list of tools that could be used after segmentation of 3D images using CobWeb.

**Thanks for pointing, we will update the figure and cite the reference in the revised manuscript.**

**Revised**

Moreover, you could compute permeabilities for your segmentations and compare then against available data, e.g., "Laboratory measurements of porosity and permeability reported by (Andrä et al., 2013b) are around 21 % ($\varphi$ = 0.21) and $\kappa$ = 150 mD ⣠Tˇ A 470 mD, respectively".

**Thanks for your suggestion, we will implement this in future studies.**

**Not revised**

CobWeb 1.0 License statement and permissions for CobWeb 1.0 package.
Copyright (C) 1993--2019
* * *
Citation
This software is distributed free of charge.  If you use it in research
resulting in a scientific presentation or publication, the software
should be acknowledged and cited as:

Chauhan,Swarup, Sell,Kathleen, Enzmann,Frieder, Rühaak,Wolfram,Wille,
Thorsten,Sass,Ingo, Kersten, Michael.
CobWeb 1.0: machine learning tool box for tomographic imaging
Zenodo : OpenAIRE. Vol. 2018, Issue December. Genève, : CERN, 2018.
* * *
License
CobWeb 1.0 is distributed under the
"Creative Commons License Version 4.0", available at
http://creativecommons.org/licenses/by-nc-sa/4.0/
        You are not allowed:
        To reuse and redistribute the APS GmBH and Technische Universität
Darmstadt Logos

You are free:
        To Share -- copy and redistribute the material in any format
        To Adapt -- remix, transform, and build upon the material

Under the following conditions:
Attribution -- You must give appropriate credit, provide a link to
the license, and indicate if changes were made. You may do so in
any reasonable manner, but not in any way that suggests the
licensor endorses you or your use.

        Non-commercial -- You may not use the material for commercial
            purposes.

        Share Alike --  If you remix, transform, or build upon the
            material, you must distribute your contributions under the same
            license as the original.

    See the above link for the full text of the license.
* * *
    Disclaimer

    This software is provided 'as-is', without any express or implied
    warranty. In no event will the author(s) be held liable for any damages
    arising from the use of this software.
* * *

[revised manuscript text omitted]

---

## Author Response (AR2)

Dear Dr. Thomas Poulet,

Today, I have uploaded along with the replies to the referees comments and the second revised version of our manuscript gmd-2018-335 title "CobWeb 1.0: Machine Learning Tool Box for Tomographic Imaging". We think these updates has helped us to further enhance the quality of the manuscript.

In the manuscript, the updated passages are colored in blue and the deleted lines are strikethrough in red.

Best Regards

Swarup Chauhan

We thank the referee for his time, his encouragement w.r.t CobWeb 1.0 and his honest and critical comments. We have done our best to address most of the comments and acknowledge the shortcoming. This has certainly improved the manuscript. Please find below a point-by-point response to your concerns.

Sincerely,

Swarup Chauhan, Kathleen Sell, Wolfram Rühaak, Thorsten Wille and Ingo Sass.

The reviewer comments are formatted in italics and the author's response to the comments are formatted in bold.

Notation RC2.P# represents Reviewers Comment. Paragraph Number

*A free, user-friendly toolbox for 3D image processing of X-ray CT imagery of porous rock and sediment is a valuable contribution to the community. The work put into this Cobweb project should definitively be rewarded with a stand-alone paper in GMD that can be cited, whenever the toolbox is used in upcoming projects. This manuscript has already undergone one round of referee comments and revisions. The revised manuscript can still be improved on several occasions, but in general I agree with the present structure of the paper. Also the Cobweb toolbox itself can be improved in many of the routines, which I will list below, but it would be too harsh to reject the paper for that. I would still suggest on more round of revisions to at least discuss these shortcomings more explicitly and also remove grammar/spelling mistakes.*

**Thanks for the appreciation.**

Specific comments:
RC2.P1. *I would tone down the novelty aspect of your dual filtering and dual segmentation approach for the gas hydrate data set (e.g. in abstract, P17L3, etc.) To me it rather sounds like a drawback to first filter with an anisotropic diffusion (AD) filter and have unsatisfactory noise removal results so that another non-local means (NLM) filter is required (or vice versa that NLM, which should also be edge-preserving when the parameters are set properly, apparently cannot do a good job without preconditioning with AD). To me dual filtering doesn't sound like something to aspire, but more like extra time required to adjust a larger set of parameters for a satisfactory result. Same holds for dual segmentation. First you have to run unsupervised K-means with many classes, only to regroup them by indexing into meaningful material classes through user interaction by an expert later on.*

**Revised page 17 line 16 -28**

**Several attempts were made to remove the edge enhancement effect (ED) using single filters and in combination with supervised techniques. But they did not yield desirable results.**

**The ED pixels values where in very close vicinity to the Methane Hydrate pixels. Therefore, preprocessing with single filters despite using appropriate settings could not normalize ED to a reasonable range of values (high STDV). So, despite tailoring customized training dataset using a representative slice– due to large stdv of ED values, methane was systematically misclassified as ED as the pixel values deviated away from trained model.**

An option was to create different training dataset using several representative slices, and introduce the unknow stack of data for classification in batches of 100 slices. This regularization trick for us did not represent a good norm for supervised ML classification.

Hence, through the experience gained in (Sell et al., 2016) for us dual-filtration was one of the best approaches we could include in preprocessing step. This dual-filtering did not removed the ED completely rather normalize it to a reasonable range. Through the approach of rescaling and (hard) K-means segmentation (dual-segmentation) we were absolutely sure that the ED artifact have been removed.

*RC2.P2. I'm not happy with this paragraph on FCM (P6L15-24). First of all, it is hard to follow for a reader that is not already familiar with k-means and fuzzy c-means. Secondly, your basically describing that FCM is incapable of dealing with partial volume voxels at material boundaries being misclassified to the intermediate class (and thus resulting in a too low volume fraction of the darkest class, i.e. porosity), since FCM is only operating in a feature space, i.e. the histogram, and cannot account for spatial features, i.e. partial volume voxels sitting on an edge vs. real intermediate material patches. So all that you can do is to tweak the FCM settings such that the partial volume problems disappear but so do the real intermediate material voxels elsewhere in the image. This drawback seems to be carefully neglected in this paragraph. Maybe remove this paragraph and replaced it with a more general statement, why FCM can be superior to KM.*

**Revised; Page 6 line 16-20**

*RC2.P3: It is next to impossible to digest this paragraph on SVM without prior knowledge (P7L19-27).*

**The explanation given in P7L19-27 is to give the reader an intuitive idea how SVM performs and to restrain the mathematical formulation which is available in the literature; supporting literature has been cited (Suykens and Vandewalle, 1999; van Gestel et al., 2004; Bishop, 2006; Haykin, 1995) (T. M. Cover, 1965) to fulfil the gap in prior knowledge, mathematical formulation and historical significance.**

*RC2.P3: Take the first sentence, for instance. How can a training dataset be non-linear, what does that actually mean? The training dataset would be a set of gray values that you obtained by clicking into the image and assigning those locations to a certain material class. Those gray values make up a 1D frequency distributions for each material class that can have substantial overlap. Where does the second dimension or even third dimension come from, that help to remove this overlap? See, I'm not even sure if this 2D or 3D coordinate system and the associated hyperplanes are in a feature space or in the spatial domain of the XCT image. Probably, I'm on the wrong track here, but so will be most of the readers.*

**A dataset can be linearly separable if the points in the dataset can be partitioned into two classes using a threshold function (threshold should not be a piecewise discontinuous function). Loosely speaking the threshold function fits a line to produce the partition.**

**Now, as pointed out by the reviewer– the training vector is a 1D array of pixels which can have substantial overlap. If I fit a threshold function to this substantially overlapped dataset– this usually leads to wrong partitioning (Bishop, 2006; Haykin, 1995). So, such dataset is regarded as linearly in-separable *alias* non-linear separable dataset (Bishop, 2006). This is what we mean as non-linear training dataset.**

**Here 3D implies a 2+1 dimensional space which consists of two spatial dimensions that correspond to the coordinated of the pixels position in the image and the third dimension to that of the greyscales that evolves as a result of the machine learning. Since we are having continuous values of the greyscales values we require a sufficiently smooth threshold to make the classification.**

*RC2.P4: The training is pretty restrictive (P12L1-21). If I understand correctly, you can only click once for one material and the class statistics is constructed from 6x6 pixels around the coordinate, where you clicked. My experience with Ilastik, https://www.ilastik.org/, another free machine-learning based segmentation toolkit, is that you can draw multiple lines of any thickness for each material and all covered pixels/voxels contribute to the class statistics. In addition, a whole set of samples can be segmented at once by only drawing training data in a small number of samples (even in live mode, i.e. the segmentation results are updated on the fly with any additional line. I think it's unfair to criticize this somewhat inflexible training mode in Cobweb. Please take this as an encouragement for further development and add Ilastik to the software survey in the beginning.*

**We admit, that CobWeb is a bit inflexible and rudimentary in terms of certain features compared to software like Ilastik. In the future versions we can improve these features.**

**We have added Ilastik to the survey of open source software's. Figure 1b. page 28**

*RC2.P5: I do not understand the paragraph on 2D slice-by-slice segmentation (P17L14-26)). Do different area fractions of each material (i.e. spatial variability of the rock) or vertical intensity variations (due to hardware shortcomings) mess up the slice-by-slice approach? If it is the former, you need to explain why different area fractions in each slice (e.g. change in porosity) affect the segmentation results, if the average gray value of pores, rocks and matrix does not change. Also, calling the Z coordinate direction in XCT data "temporal information" is very strange.*

**Revised, Page 18 -Line 8 – 22**

**We thank the reviewer for his advice, we changed this part of the text to specify the issue more precisely. In short the topic discussed is only related to the fact that processing is done in 2D and the results are later on combined to a 3D result. Since each slice is an snapshot of the total volume fraction of different materials– by performing slice-by-slice segmentation one gets only the partial information (change in porosity) of the total volume fraction of the respective materials. Therefore, averaging helps to obtain the 3D information. Although this approach is favorable in terms of performance it has shortcomings in terms of possible artefacts.**

*RC2.P6: Chapter 5.3 on Multi-Phase Segmentation: Is this section for all three datasets? If so, what is actually the third material beside pores and rock in Berea sandstone (and Grosmont carbonate rock)? What does it mean that the intermediate class has a Poisson distribution (P18L9)?*

*It is a bit discouraging to read that the supervised methods did not result in better segmentations then unsupervised K-means (which has been around for many decades and needs to be cleaned up with your supervised dual segmentation strategy in one out of three show case datasets). So my take-home message is that you 'sell' Cobweb as the first ML-only segmentation toolbox for multi-phase segmentation (P1-19-20 in abstract), only to use K-means throughout the paper which is essentially available in all other commercial or non-commercial toolboxes. I think that this is a shot in the foot,*

*especially since one reason against using the other ML methods in Cobweb is that they are apparently too slow at the moment. Don't you think it would be better to show the LSSVM or ensemble classifier results instead?*

**Not revised.**

**We acknowledge the reviewer's concern. We think within the scope of the paper which introduces CobWeb and edge enhancement segmentation, the detailed verification with LSSVM and ensemble classifiers has been undermined to some extent. But, the previous work Chauhan et al., (2016) based on which the CobWeb is developed; benchmarks different ML algorithms and draws suitable conclusions. Therefore, this gives us the confidence that CobWeb is usable and durable.**

*RC2.P7. More info on the watershed method is required (P18L31-P19L9). The x-axis in Fig. 6c (pore radius) suggests a maximum inscribed sphere method to me, but the traditional watershed transform on binary data creates irregular shaped fragments. How can an irregularly shaped object have a single pore radius?*

**Revised Page 19 to page 20 line 1**

**We have added some more information on the watershed method as suggested by the reviewer. No, the algorithm does not use inscribed sphere method but the traditional watershed transformation as implemented in Rabbani et al., (2014). and assumes the irregular shape as spherical pores.**

*8. You simply have not reached an REV for PSD histograms within a single slice (P19L12). My educated guess is that PSD requires an even larger REV than porosity, and yet (against your own advice) you show the PSD of each individual slice in an overloaded figure instead of a single PSD for the entire 3D REV. What do you learn from such a figure? This needs to be changed.*

**Not revised.**

**Yes, we accepted this as a short coming of the current research work the implementation will be corrected in the future versions. In the discussion (Page 17-18 | 28-5) addressed this issue and averaging helps to obtain the 3D information. Although this approach is favorable in terms of performance it has shortcomings in terms of possible artefacts.**

***Technical comments:***
*P2L31: tackle with this -> tackle this*

**Changed.  Page 2 line 31**

*P3L6: Why is fspecial striked through?*

**Corrected.**

*P3L9: This sentence sounds incomplete. Remove 'Despite'?*

**The sentence has been slightly changed, the changes are marked in blue in the text.**

**Page 2 line 11 -line 12**

*P5L4: masking in -> filtering is - Masking is the wrong word here. You convolve the image with a filter kernel (or just apply a filter). Also, I couldn't follow why two Laplace filters are required and how exactly they are implemented. Is it a Laplace-of-Gaussian with two different sigmas, first a large Gaussian sigma for thick edges followed by a lower Gaussian sigma for thin edges? Is the second applied to the result of the first (would make no sense, since all small features are gone) or to the original image and the outcome somehow combined with the outcome of the first?*

**Revised: Page4 line 28 – page 5 line 2**

**Obviously, as pointed by the reviewer, irrespective of large sigma or small sigma first, it will lose all the features.**

**Within the fspeacial routine we use only the averaging filter we have not implemented the Laplacian or sobel filter. Therefore, we do not do any sequential filtering (repeated) on the same image. Within the for-loop each slice gets filtered only once and later are stacked together.**

*P5L9: Remove 'Whereas,'*

**Deleted**

*P5L20: comprises of pixels -> comprises pixels*

**Changed. Page 5 line 20**

*P7L15: unknow -> unknown*

**Changed. Page 7 line 22**

*P6L30: models -> model's*

**Changed. Page 7 line 6**

*P7L19: Remove 'Now,'*

**Changed. Page 7 line 26**

*P9L3: atleast -> at least*

**Changed. Page 9 line 10**

*P9L22: Mix of present tense and past tense.*

**modified. Page 10 line 1-2**

**The accuracy is determined by calculation area under the curve (AUC), and the simplest**  **way to do this was by using trapezoidal approximation**.

*P10L21: where -> were*

**Changed. Page 10 line 28**

*P11L13: Meaning of densely nested function unclear*

**Changed. Page 11 line 20**

*P11L23: The term 'back-end' might be uncommon outside the computer science world.*

**Page 11 line 32**

**Changed to at the data access layer (also know as back-end)**

*P11L25-26: Combine the two sentences into one*

**Changed. Page 12 line 1-3**

**It is an easy, one-step process in the case of unsupervised techniques i.e. based on the options selected in the preprocessing** *uitable***, the image is filtered and subsequently, segmented.**

*P11L26: addition -> additional*

**Changed. Page 12 line 3**

*P13L10: The beginning of chapter 4 is rather abrupt. I would write a short summary like: "this concludes the description of the Graphical user interface. For more information the user manual can be consulted which is available as supporting information. In the following Cobweb toolbox is demonstrated by means of three showcase examples, which are briefly introduced in terms of underlying research question, imaging settings and challenges for image processing." - or something similar*

**Thanks for the suggestion. Following sentences have been added. Page 13 line 22 -line 26**

**This concludes the description of the section toolbox and functionalities. For more information on the usage of the graphical user interface the user manual can be consulted, which is available as supporting information.**

**In the following, sections the CobWeb toolbox is demonstrated by means of three showcase examples, which are briefly introduced in terms of underlying imaging settings, research question and challenges for image processing.**

*P13L22: Write out ED at first occurrence of the abbreviation*

**For consistency the abbreviation ED has been remove from the manuscript**

*P15L8: Remove 'Now,'*

**Removed. Page 15 line 22**

*P15L28 10243 ->1024^3*

**Done. Page 16 line 10**

*P18L9: Do you mean really mean 'low sample size' or 'low volume fraction'?*

*This is not necessarily the case. A counter example would by a mud-rock with a large volume fraction of the matrix (intermediate class) with low porosity and a few inherent dense rocks.*

**Changed, page 19 line 4**

**Yes. low volume fraction is the appropriate word. The wording has been replaced in the manuscript.**

*P19L4: Explanation is required, why it is so much lower than the porosity values given previously in the material description (mostly due to sub-resolution pores, I guess).*

**Thanks for the pointers. Following explanation has been added to page 20 line 3 - 6**

**Particularly, in the case of Grosmount, after segmentation the obtained porosity value $\varphi$ =10.5 ± 2.3 % is extremely low compared to the laboratory measurement $\varphi$ =21 % published in (Andrä et al., 2013). Exact reason is not known but could also be partly attributed to sub-resolution pores which couldn't be captured do to low resolution obtained through XCT measurement.**

*P19L29: These discrepancies in modelling and transport simulation have not been addressed in the main paper. Thereof it is not appropriate to mention them all of a sudden in the conclusions.*

**It has been removed. Page 20 | L30**

*Fig3: What do the different blue colors for different boxes stand for?*

**The figure has been rectified and caption has been updated. Page 30**

*Fig5: The color bar is misleading. There are only three materials and not five. You might need to create your own color legend if the built-in matlab functionality is not flexible enough to do that. I'm not sure what rescaled (Fig. b) means in this context. Should be explained in the caption.*

**The color bar has not been changed. The caption has been modified as follows. Page 32**

**Figure 1: 2D slices of REV 1 are represented above. The raw image is first filtered with anisotropic diffusion filtered and later on with non-local means. Thereafter, the different phases where segregated using a segmentation and indexing approach and the raw image(s) is rescaled such that they aren't any overlap or mixed phases within the raw image; and example is shown as the rescaled 2D ROI plot. Thereafter K-means segmentation is performed on the complete stack; 2D images of slice 1, slice 20 and slice 695 are shown as examples.**

Dear Dr Kirill Gerke,

Thanks. We have added the references and updated the Figure 1.

Sincerely,

Swarup Chauhan, Kathleen Sell, Wolfram Rühaak, Thorsten Wille and Ingo Sass.

Notation *RC1.P#* represents Reviewers Comment. Paragraph Number

I think Authors did quite a job to address the comments of all three reviewers.
It did improve the paper. While i personally would argue with a couple of statements, i think it deserves to be published and total agreement between authors and reviewers should not be a requirement.
If the software in the paper would be written in some interpretable language and not in proprietary Matlab - it would be even better then it is now.

*Just one small thing:*
*RC1.p1: I saw Authors adding FDMSS to Fig.5 and thank then for this help in reaching potential users. But i have to admit that FDMSS solves only single-phase flow problem and do not possess any image processing functionality expect for creation of 3d input image out of 2d images stack. For this it looks rather unfair to see FDMSS's "level" to be higher than the of OpenPNM, which does have a wider range of functionality...*

**The reviewer meant Fig. 1 instead of Fig. 5; Fig. 1 has been modified.**

*I also did not see FDMSS and OpenPNM's references in the reference and would for OpenPNM would suggest to cite Jeff's Gostick, J. T. (2017). Versatile and efficient pore network extraction method using marker-based watershed segmentation. Physical Review E, 96(2), 023307 as it adds the major functionality to the package.*

**Revised. Page 20 line 28-29**

*All in all, from my point of view the paper is good to go.*

[revised manuscript text omitted]

---

## Author Response (AR3)

Dear Dr. Thomas Poulet,

We thank you for your encouragement, patience and support throught this long review process. I was a good learning experience. We appreciate it.

On behalf of all the co-authors

Sincerely,

Swarup Chauhan

RC2.P3: please add clarification to the "linear" and "3D" terms in the main text.

The terms were clarified in the text.

RC2.P6: please answer reviewers specific questions ("is this section for all three datasets ? if so what is actually the third material (…)?

Yes, this section is for all the three datasets. We have added the following sentences to explain the the third material

In the case of Berea sandstone, the segmentation was restricted to four clusters out of which three phases can be clearly seen. The first two phases being pores and rock, in the third phase minerals Ankerite, Zircon, K-feldspar, Quartz, and Clay have been classified into single mono-mineral phase and the fourth phase comprise of small scale features like residual speck and noise pixels. The Grosmont carbonate sample was also segmented into four clusters, comprising of pore, pore inclusions, Calcite and brightness inhomogeneities of noise classified and the fourth phase.

Revised: Page 18 to Page 19 Line 33 to line 4

What does it mean that the intermediate class has a Poisson distribution? ")

The sentence has been rephrased and no longer mentions the Poisson distribution.

Revised. Page 18 line 14-16

RC2.P6: While I agree that showing the LSSVM or ensemble classifier go beyond the scope of this study, please clarify your answer about the comment regarding the statement P1 L19-20 in the abstract. State explicitly the " suitable conclusions" from your previous if they address the comment directly, and/or adjust any phrasing in the text where required to address the reviewer's point.

We have rephrased the text in the abstract and added new text in section 5.3 .

We have added the following sentence to the abstract

In this study, we demonstrate image segmentation using unsupervised machine learning techniques.

Page 1 line 19-20.

In the Section 5.3 following text has been added.

Note that the emphasis of this study was to demonstrate the capabilities of CobWeb and removal of edge enhancement segmentation, through dual filtration and dual segmentation schemes. Detailed verification with LSSVM and ensemble classifiers falls therefore outside the scope of this work and readers are referred to the previous work from (Chauhan et al., 2016a) based on which CobWeb is developed. That work benchmarks different ML algorithms and quantifies their respective accuracies and performances.

Page 19 line 6 -10

RC2P8: I don't think that the reviewer is asking for any improvement of the code, but pointing out that the individual slices are not large enough to be respesentative (RVE issue), which I can´t see discussed pages 17-18 L28-5. Please address this comments.

We have revised the text. Following sentence has been added to the text.

[revised manuscript text omitted]